# Replication stress triggers microsatellite destabilization and hypermutation leading to clonal expansion in vitro

Yusuke Matsuno[1,2], Yuko Atsumi[1], Atsuhiro Shimizu[1], Kotoe Katayama [3], Haruka Fujimori[1,4], Mai Hyodo[1,4], Yusuke Minakawa[1,4], Yoshimichi Nakatsu[5], Syuzo Kaneko[6], Ryuji Hamamoto[6,7], Teppei Shimamura[8], Satoru Miyano[3], Teruhisa Tsuzuki[5], Fumio Hanaoka[9,10] & Ken-ichi Yoshioka[1]

Mismatch repair (MMR)-deficient cancers are characterized by microsatellite instability (MSI) and hypermutation. However, it remains unclear how MSI and hypermutation arise and contribute to cancer development. Here, we show that MSI and hypermutation are triggered by replication stress in an MMR-deficient background, enabling clonal expansion of cells harboring ARF/p53-module mutations and cells that are resistant to the anti-cancer drug camptothecin. While replication stress-associated DNA double-strand breaks (DSBs) caused chromosomal instability (CIN) in an MMR-proficient background, they induced MSI with concomitant suppression of CIN via a PARP-mediated repair pathway in an MMR-deficient background. This was associated with the induction of mutations, including cancer-driver mutations in the ARF/p53 module, via chromosomal deletions and base substitutions. Immortalization of MMR-deficient mouse embryonic fibroblasts (MEFs) in association with ARF/p53-module mutations was ~60-fold more efficient than that of wild-type MEFs. Thus, replication stress-triggered MSI and hypermutation efficiently lead to clonal expansion of cells with abrogated defense systems.

[1] Division of Carcinogenesis and Cancer Prevention, National Cancer Center Research Institute, Tsukiji, Chuo-ku, Tokyo 104-0045, Japan. [2] Department of Applied Chemistry, Faculty of Science, Tokyo University of Science, Kagurazaka, Shinjuku-ku, Tokyo 162-8601, Japan. [3] Human Genome Center, Institute of Medical Science, University of Tokyo, Shirokanedai, Minato-ku, Tokyo 108-8639, Japan. [4] Biological Science and Technology, Tokyo University of Science, Niijuku, Katsushika-ku, Tokyo 125-8585, Japan. [5] Department of Medical Biophysics and Radiation Biology, Faculty of Medical Sciences, Kyushu University, Maidashi, Higashi-ku, Fukuoka 812-8582, Japan. [6] Division of Molecular Modification and Cancer Biology, National Cancer Center Research Institute, Tsukiji, Chuo-ku, Tokyo 104-0045, Japan. [7] Cancer Translational Research Team, RIKEN Center for Advanced Intelligence Project, Chuo-ku, Tokyo 103-0027, Japan. [8] Division of Systems Biology, Graduate School of Medicine, Nagoya University, Tsurumai-cho, Syouwa-kuNagoya 466-8550, Japan. [9] Faculty of Science, Gakushuin University, Mejiro, Toshima-ku, Tokyo 171-8588, Japan. [10] National Institute of Genetics, Mishima, Shizuoka 411-8540, Japan. Correspondence and requests for materials should be addressed to K.-i.Y. (email: kyoshiok@ncc.go.jp)

Most cancers develop in association with mutations and genomic instability, e.g., chromosomal instability (CIN) or microsatellite instability (MSI)[1]. CIN encompasses a wide variety of chromosomal abnormalities, including chromosomal rearrangements and aneuploidy[2,3], whereas MSI is defined as changes in the lengths of microsatellite fragments that contain short repetitive sequences (1–6 bases)[4]. Mismatch repair (MMR) status is a major determinant of whether CIN or MSI is induced[5]: MSI develops in MMR-deficient cancers, including those in which MutSα (MSH2/MSH6 complex) and MutLα (MLH1/PMS2 complex) are mutated[6,7]. Cancer cells with MSI generally exhibit hypermutation[8–10]. Importantly, such genomic destabilization promotes cancer development, e.g., mutations in the breast cancer susceptibility genes BRCA1 and BRCA2 can cause CIN, leading to the development of cancer[11,12].

During the initial stages of cancer development, cells often accumulate DNA replication stress-associated DNA double-strand breaks (DSBs) and develop genomic instability[13–15]. Cultured cells exhibit the same phenotypes upon oncogene activation and exposure to exogenous growth stimuli; for example, CIN is induced due to the accumulation of replication stress-associated DSBs[14,16]. The importance of CIN induction in cancer development is probably related to the associated induction of cancer-driver mutations, as suggested by a study of MMR-proficient mouse embryonic fibroblasts (MEFs), which showed that immortalization associated with ARF/p53-module mutations[17] is blocked unless CIN is induced[18,19]. MSI is prominent in MMR-deficient backgrounds. Although chromosomal abnormalities are also observed in MMR-deficient cancer cells[20,21], the level of CIN is usually much lower than that in MMR-proficient cancer cells[1]. It remains unclear whether CIN and MSI are mechanistically related.

Mismatches that arise during DNA replication are corrected by MMR[22,23]. The MMR proteins associate with the replication fork by interacting with proliferating cell nuclear antigen (PCNA)[24,25]. In MMR-deficient cells, mutations accumulate during canonical replication[26,27]. This is generally thought to increase the risk of cancer-driver mutations[8–10], but it raises the question of whether MSI is induced in association with the accumulation of replication errors in an actively replicating state, or is instead induced as an alternative to CIN in response to replication stress-associated DSBs in senescent cells. In addition, it remains to be determined whether MSI is associated with the induction of cancer-driver mutations.

This study investigated the mechanisms via which hypermutation and ARF/p53-module mutations occur, and how MSI is induced. Our results revealed that replication stress-associated DSBs induce MSI in MMR-deficient cells while CIN is suppressed. Hypermutation also arose during this process, leading to clonal expansion of cells with abrogated defense systems, including those with ARF/p53-module mutations.

## Results

**MSI in MMR-deficient cells as an alternative to CIN.** To explore the mechanisms via which MSI and mutations are induced, we compared the immortalization of MMR-deficient ($Msh2^{-/-}$) MEFs with that of MMR-proficient ($Msh2^{+/+}$) MEFs. Wild-type MEFs usually immortalize with ARF/p53-module mutations[17,19] and CIN (tetraploidy)[16,18], as in MMR-proficient cancer cells. As previously reported for $Msh2^{+/+}$ MEFs[16,18], $Msh2^{-/-}$ MEFs progressed to a senescent state and subsequently immortalized under the standard 3T3 (Std-3T3) protocol (Fig. 1a). As expected based on the genomic instability phenotypes of cancer cells, $Msh2^{+/+}$ MEFs immortalized with CIN (tetraploidy) but without MSI, whereas $Msh2^{-/-}$ MEFs immortalized with stable diploid

and MSI (Fig. 1b–d; Supplementary Fig. 1a–e; see red arrows showing the signal change corresponding to MSI induction). Although CIN (tetraploidy) and MSI were induced in a mutually exclusive manner, they were not completely distinct: chromosomal abnormality induction was suppressed, but still detectable, even in $Msh2^{-/-}$ MEFs (Supplementary Fig. 2). This is similar to the situation in MMR-deficient cancer cells, in which CIN-associated chromosomal abnormalities are generally suppressed but not completely blocked[1]. In addition, we observed an MSI-associated peak shift only at D17mit123 (Fig. 1) or at D17mit123 and D7mit91 (Supplementary Fig. 1e), but not at other loci. This suggests that there are hotspots of MSI but that each of these loci is not always destabilized, similar to the situation for CIN-associated genomic rearrangements.

To determine when MSI is induced, we monitored MSI status during the immortalization process of $Msh2^{-/-}$ MEFs. We observed an MSI-associated secondary peak at P8 + 15 days (Fig. 1e), the same time at which CIN is induced in $Msh2^{+/+}$ MEFs[16,18]. $Msh2^{+/+}$ MEFs at this time point are vulnerable to replication stress caused by continuous exposure to growth stimuli. Consequently, CIN is induced under the Std-3T3 protocol[16,18], but not under a temporary serum-depleted-3T3 (tSD-3T3) protocol that does not involve continuous growth stimulation, in which immortalization is blocked[18]. Therefore, we cultivated $Msh2^{-/-}$ MEFs under the tSD-3T3 protocol. As expected, these MEFs retained genome stability and did not immortalize (Fig. 2a), indicating that MSI is also induced upon continuous exposure to growth stimuli. Thus, MSI in $Msh2^{-/-}$ MEFs is induced as an alternative to CIN in $Msh2^{+/+}$ MEFs that undergo immortalization during continuous exposure to growth stimuli. Given that replication stress-associated DSBs trigger CIN[16], these results suggest that replication stress is also involved in MSI induction under the $Msh2^{-/-}$ background.

**Replication stress-associated MSI induction.** To directly study replication stress status, we monitored γH2AX foci as a marker of damage that could arise in cells treated as shown in the workflow (Fig. 2b). As expected, many γH2AX foci were observed in EdU-positive cells (Fig. 2b), suggesting that those damages were caused during the replication. The γH2AX foci induced under these conditions were mostly colocalized with 53BP1 foci, suggesting that they represented DSBs. ssDNA that is exposed in association with replication stress is detectable as foci of pre-incorporated BrdU under native conditions[28–30]. Accordingly, we monitored BrdU foci in MEFs treated as shown in the workflow depicted in Fig. 2c. γH2AX and 53BP1 foci mostly colocalized with BrdU foci (Fig. 2c and Supplementary Fig. 3a–c). These results support the idea that DSBs arose in association with replication stress. Further evidence that MEFs were under replication stress in this condition was provided by monitoring the status of phosphorylated RPA and ATR (Supplementary Fig. 3d). Collectively, these findings show that MSI is induced when replication stress-associated DSBs accumulate, as previously shown for CIN[16,18]. EdU-negative MEFs cultured under the tSD-3T3 protocol contained significantly fewer γH2AX foci after 72 hr than those cultured under the Std-3T3 protocol (Fig. 2b), consistent with the genome stability phenotypes. Under the latter protocol, the number of γH2AX foci was significantly lower in EdU-negative $Msh2^{-/-}$ MEFs than that in EdU-negative $Msh2^{+/+}$ MEFs (Fig. 2b). This difference suggests that the repair efficiency of DSBs differs according to the MMR status, which may be associated with the relative likelihood of induction of CIN vs. MSI.

To compare the effects of CIN and MSI induction, we independently subcultured $Msh2^{+/+}$ and $Msh2^{-/-}$ MEFs at P8 at the indicated densities, and monitored them as they underwent

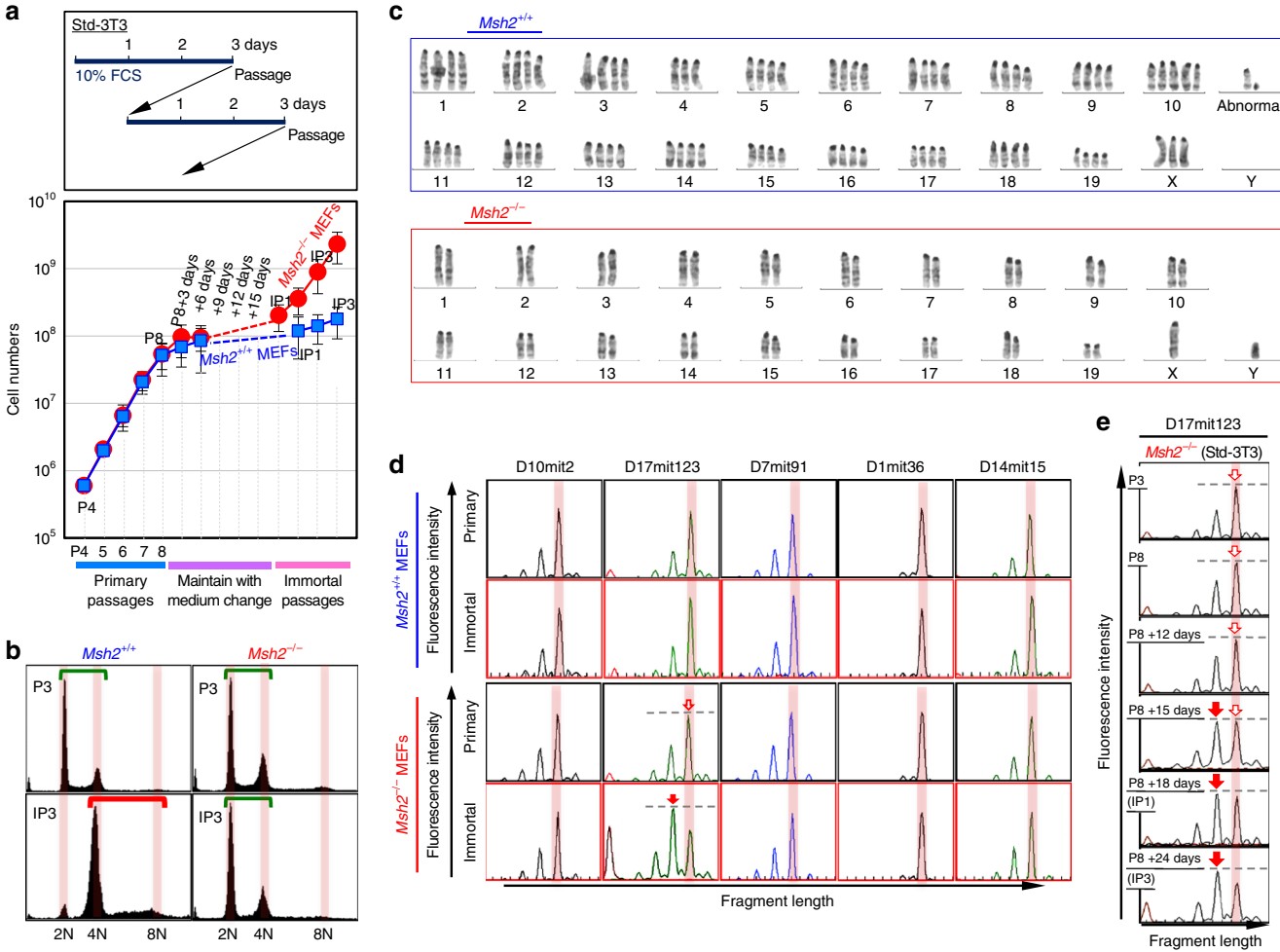

**Fig. 1** MSI is induced as an alternative to CIN in MMR-deficient cells. **a** $Msh2^{+/+}$ and $Msh2^{-/-}$ MEFs were cultivated under the Std-3T3 protocol to monitor the immortalization process. The graph shows mean cell numbers ± s.d. ($n = 3$ independent experiments with MEFs prepared from independent fetuses). **b**, **c** CIN-induction statuses were determined by flow cytometry (**b**) and G-band analysis (**c**). Green and red bars in **b** indicate diploidy and tetraploidy, respectively. **d**, **e** MSI statuses were determined by comparing fragment lengths at the indicated loci (**d**). The MSI status was determined in $Msh2^{-/-}$ MEFs at each stage (**e**). Red arrows indicate the shifted fragment peaks, i.e., MSI

immortalization. $Msh2^{-/-}$ MEFs immortalized much faster than $Msh2^{+/+}$ MEFs (Fig. 2d). The immortalization frequency of $Msh2^{-/-}$ MEFs was about 60-fold higher than that of $Msh2^{+/+}$ MEFs (Fig. 2e). Given that MEFs immortalize with ATR/p53-module mutations, these findings suggest that the risk of cancer-driver mutations is particularly high when $Msh2^{-/-}$ MEFs are subjected to replication stress, which induces MSI.

**MSI-associated hypermutation in MMR-deficient cells**. To investigate mutation induction, whole-exome sequencing was performed of MEFs at different stages (Fig. 3a: P4, P8, IP1, and IP28). Hypermutation was detected in $Msh2^{-/-}$ MEFs at IP1 and IP28 (Fig. 3b; Supplementary Fig. 3d). These mutations were mainly insertions/deletions and base substitutions inducible with G/T-mismatches (Fig. 3b, c), reflecting MMR deficiency[8,23]. About 89% of mutations detected at IP1 were observed in 30–60% of total reads (Fig. 3d). Most were carried over into IP28 (Supplementary Fig. 3e) and were then detected in 40–60% of total reads. This indicates that immortalized $Msh2^{-/-}$ MEFs with detectable mutations were clones of a single, immortalized, diploid MEF in which mutations were mostly induced in one allele. In fact, immortalized MEFs usually formed a colony (Supplementary Fig. 3f) and subsequently became dominant.

Similar results were obtained using $Msh2^{+/+}$ MEFs, although the majority of mutations observed at IP1 were detected in <40% of the total read (Fig. 3d), which probably reflects their tetraploid state (Fig. 1b, c). These cells eventually became aneuploid and therefore their chromosome status changed[18]. Consequently, many of the mutations carried over into IP28 were detected in more than 40% of total reads. These results support the idea that a single immortalized MEF underwent clonal expansion.

In $Msh2^{-/-}$ MEFs, the number of base substitutions detected at IP1 in total exons was significantly higher than expected based on errors that could arise during normal replication in a single MMR-deficient cell (Fig. 3e). These include replication errors that accumulate from embryogenesis and are not detected at P4 or P8, but become detectable after clonal expansion during immortalization at IP1. This unexpectedly high level of mutations implies that they are induced via another pathway. The number of mutations was further increased at IP28 (Fig. 3b, c). Most of these likely did not arise during IP1–IP28 because such mutations did not clonally accumulate and hence were not detectable (Supplementary Fig. 3g, h). Instead, mutations detected at IP28 included those that arose during P8–IP1 but were not detected at IP1 and became detectable after expansion during IP1–IP28 (Supplementary Fig. 3i–k). Importantly, the number of mutations detected at IP28 was again significantly higher than expected based on errors

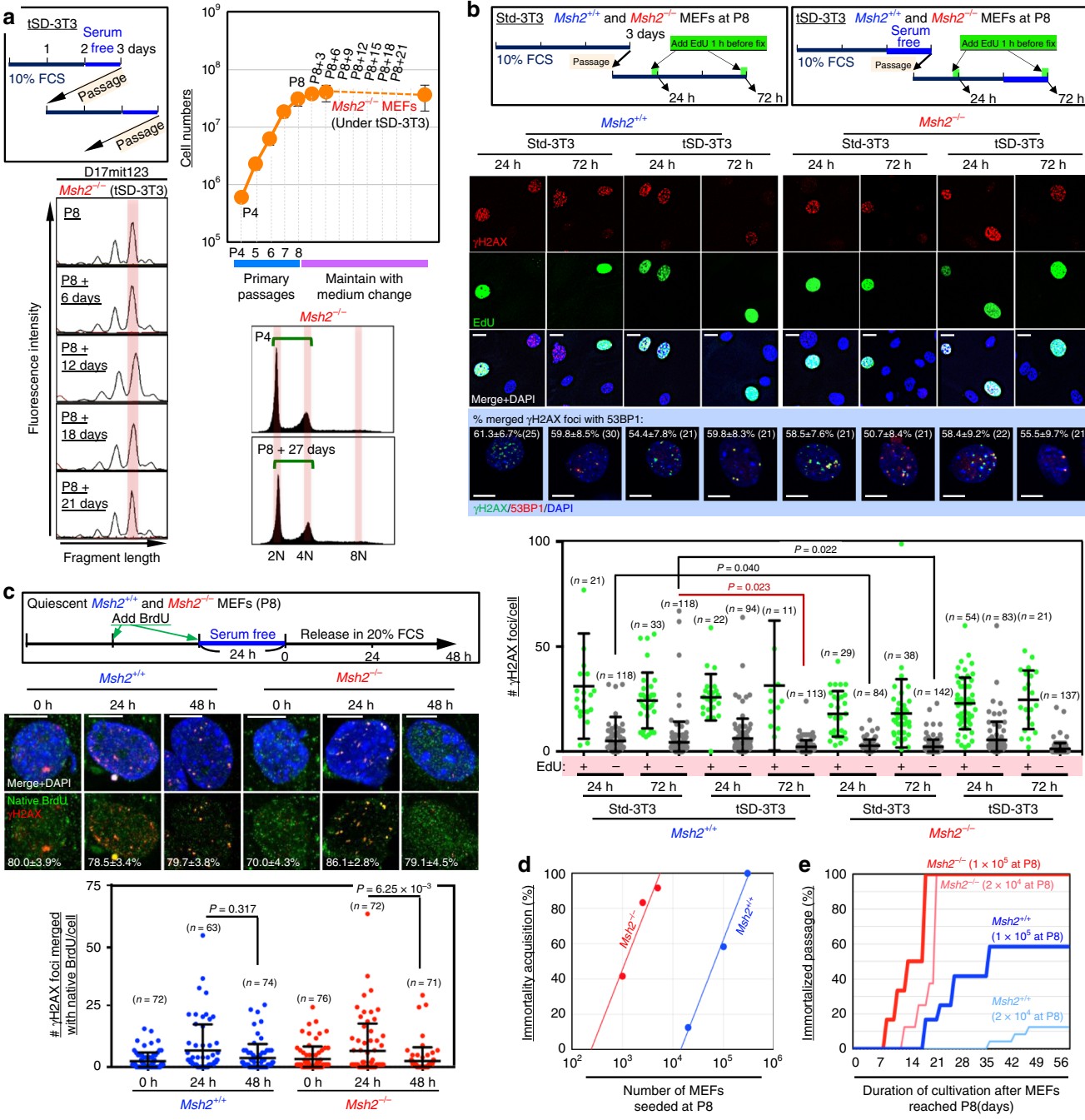

**Fig. 2** Replication stress is associated with MSI induction in MMR-deficient cells. **a** $Msh2^{-/-}$ MEFs were cultivated under the tSD-3T3 protocol, and the effects on immortalization and the MSI status were monitored. The graph shows mean cell numbers ± s.d. ($n = 3$ independent experiments with MEFs prepared from independent fetuses). **b** $Msh2^{+/+}$ and $Msh2^{-/-}$ MEFs were treated as shown in the workflows. γH2AX foci were detected by immunofluorescence with or without EdU staining. γH2AX and 53BP1 were detected by immunofluorescence ($n$ numbers are indicated in graph). Percentages of γH2AX foci merged with 53BP1 foci (means ± s.e.) are indicated in each image. Bars show means ± s.d. Scale bars, 10 μm. Two-tailed Welch's $t$-test was used for statistical analysis. **c**, $Msh2^{+/+}$ and $Msh2^{-/-}$ MEFs were treated as shown in the workflow. Foci of γH2AX and BrdU under native conditions were detected by immunofluorescence ($n$ numbers are indicated in graph). Percentages of γH2AX foci merged with native BrdU foci (means ± s.e.) are indicated in each image. Bars show means ± s.d. Scale bars, 10 μm. Two-tailed Welch's $t$-test was used for statistical analysis. **d**, **e** Multiple subcultures of $Msh2^{+/+}$ and $Msh2^{-/-}$ MEFs (P8) were seeded at the indicated cell numbers. The frequency (**d**) and speed (**e**) of immortalization were analyzed

that could arise due to replication errors (Fig. 3e), indicating that hypermutation was induced during P8–IP1. The total read depths of most mutations newly detected at IP28 were lower than those of mutations detected at IP1 (Fig. 3d), supporting the idea that they were induced during the expansion of an immortalized MEF.

**In/Del induction in repetitive loci in MMR-deficient cells.** Mutation data revealed that In/Del mutations accumulated in $Msh2^{-/-}$ MEFs (Fig. 3c), as usually seen in MMR-deficient cancer cells[31], but not in $Msh2^{+/+}$ MEFs. Importantly, these mutations were specifically induced in repetitive loci (Fig. 3f, g),

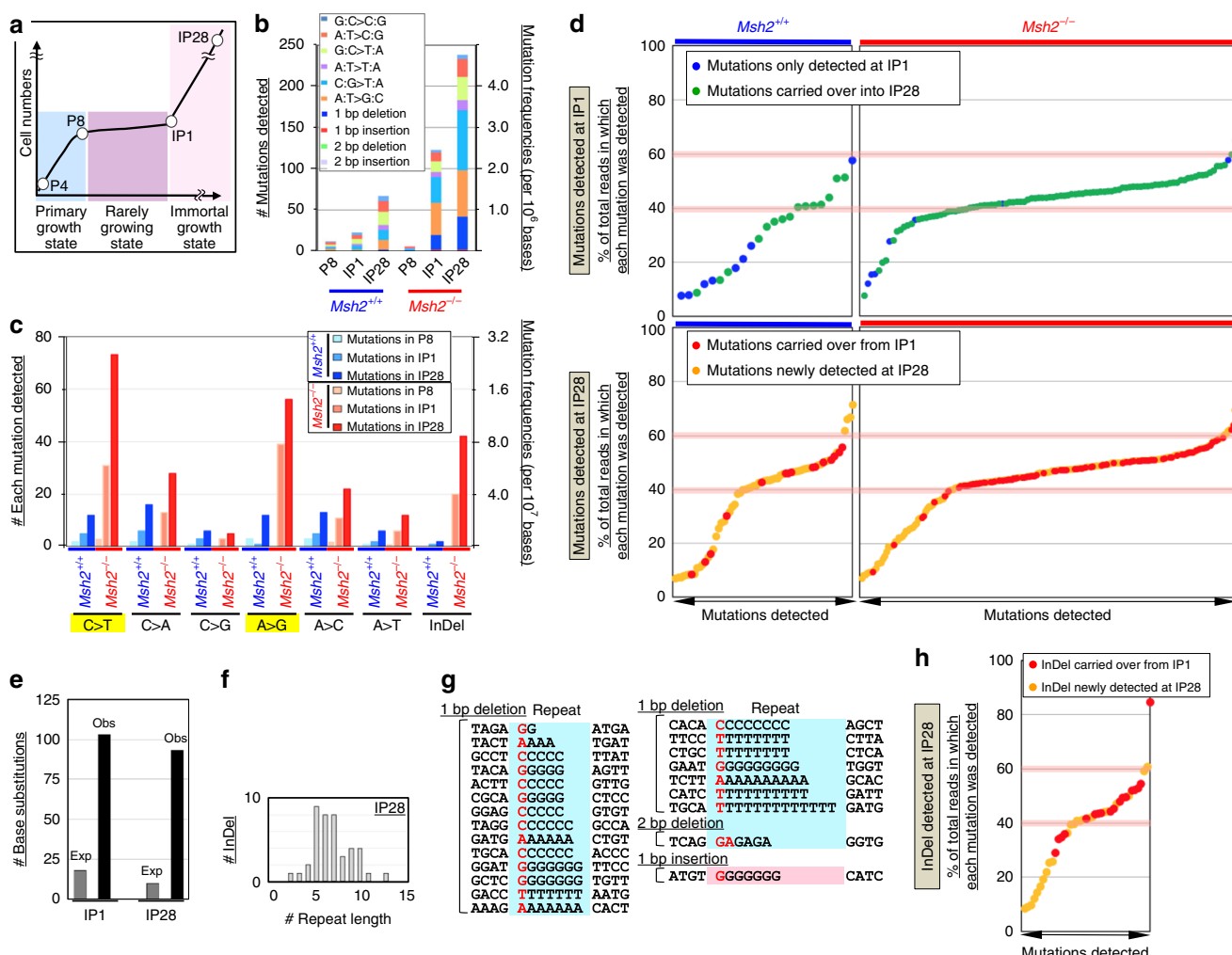

**Fig. 3** Replication stress induces hypermutation and MSI in $Msh2^{-/-}$ MEFs. **a–c** Mutations were analyzed in $Msh2^{+/+}$ and $Msh2^{-/-}$ MEFs at each cellular stage (**a**). Mutations detected in exons were categorized according to the indicated types (**b**, **c**). Mutations inducible with GT-mismatches are marked in yellow. InDel indicates In/Del mutations. **d** Mutations detected in exons of $Msh2^{+/+}$ and $Msh2^{-/-}$ MEFs at IP1 and IP28 are indicated. The percentage of total reads in which each mutation was detected is shown. **e** The numbers of base substitutions observed (Obs) in exons of $Msh2^{-/-}$ MEFs at IP1 and IP28 are indicated and compared with the numbers expected (Exp) based on replication errors. **f**, **g** In/Del mutations detected in $Msh2^{-/-}$ MEFs were analyzed. Sequences around inserted/deleted bases in immortalized $Msh2^{-/-}$ MEFs. The repeat numbers of insertions/deletions (**g**) and sequences around inserted/deleted bases in immortalized $Msh2^{-/-}$ MEFs (**h**) were analyzed. A value of 0 indicates no repeats. **h**, In/Del mutations detected in exons of $Msh2^{-/-}$ MEFs at IP28 are indicated. The percentage of total reads in which each mutation was detected is shown

indicative of MSI. Plots of the percentages of total reads in which mutations were detected yielded similar curves for In/Del mutations (Fig. 3h) and other mutations observed in $Msh2^{-/-}$ MEFs (Fig. 3d), implying that they were induced during P8–IP1 in association with replication stress-triggered MSI. This induction of In/Del mutations in repetitive loci was analogous to that observed in MMR-deficient human cancer cells (Supplementary Fig. 4a–c). In addition, In/Del mutations were more efficiently induced in loci replicating in early S phase than in loci replicating in late S phase in both immortalized MEFs and human cancer cells, supporting their relevance to MSI induction in human cancer cells (Supplementary Fig. 4d). Furthermore, base substitutions detected in exons were also more efficiently induced in loci in early S phase in both immortalized MEFs and human cancer cells with MSI, supporting the association of MSI and base-substitution inductions and their relevance to those in human cancer cells (Supplementary Fig. 4e). Analysis of these data unexpectedly revealed that In/Del mutations accumulated at particularly high levels in intronic regions at IP1 and IP28

($P < 0.001$, Supplementary Fig. 4f–l), and specifically at repetitive loci (Supplementary Fig. 4h–l). These observations imply that MSI was massively induced in intronic regions upon replication stress.

**Replication stress-triggered ARF/p53-module mutations.** We next investigated ARF/p53-module mutations, which are directly associated with MEF immortalization[16–19]. Analogous to cancer mutations that perturb ARF function[32,33], exome signals in the *Cdkn2a* gene, which encodes ARF, were almost completely absent after IP1 in $Msh2^{-/-}$ MEFs (Fig. 4a), suggesting that this locus was deleted during P8–IP1 when cells were subjected to replication stress and MSI was induced. Although chromosomal deletion is generally associated with CIN, it occurred even in $Msh2^{-/-}$ MEFs during P8–IP1, probably because some CIN-associated genomic alterations arose even in $Msh2^{-/-}$ MEFs (Supplementary Fig. 2). In fact, *Cdkn2a* was deleted in some MSI-high cancers (Supplementary Fig. 5a).

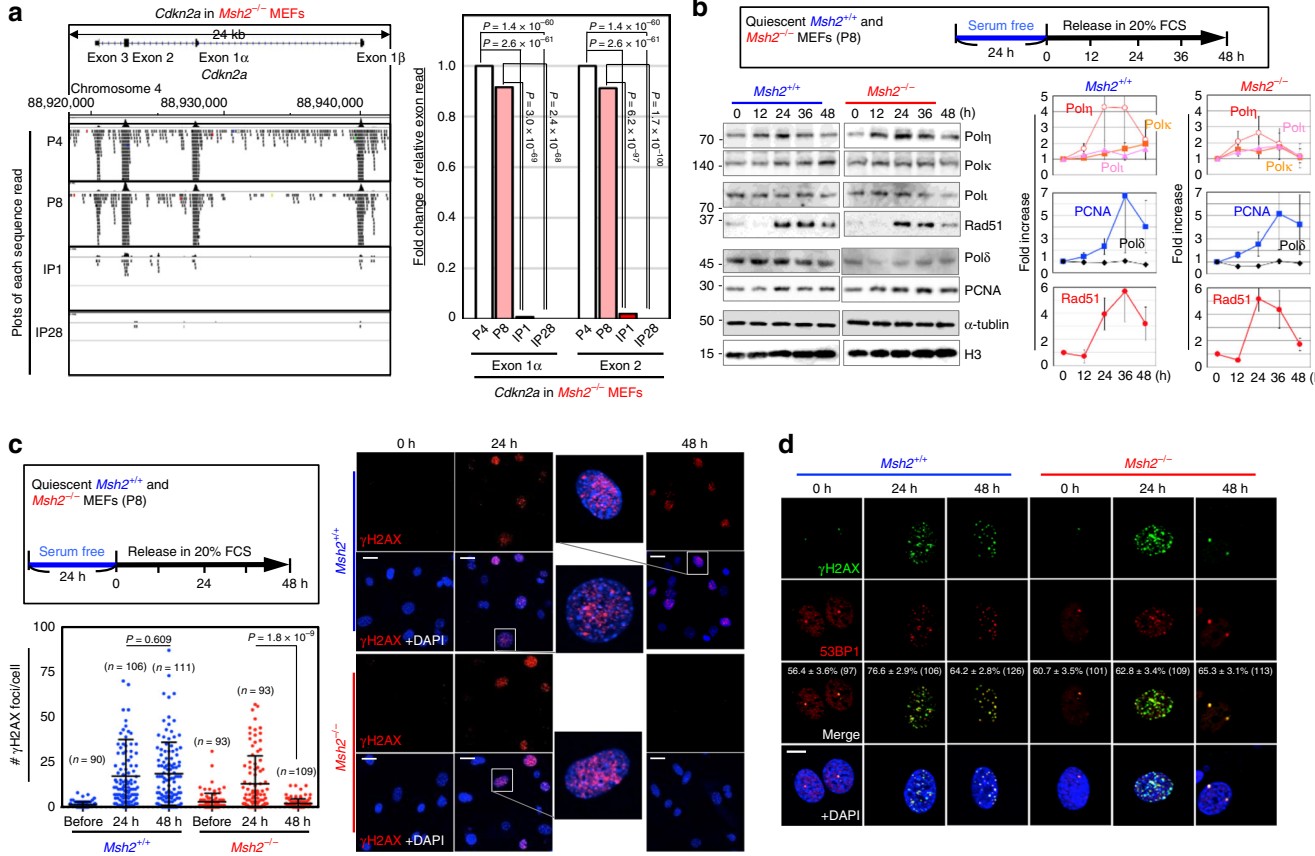

**Fig. 4** Replication stress-triggered MSI/hypermutation is associated with the induction of ARF/p53-module mutations. **a** Individual sequence reads obtained during exome analyses were aligned onto the *Cdkn2a* gene locus and visualized using the Integrative Genomics Viewer at each cellular stage. Deletion status of this gene locus was also analyzed based on numbers of sequence reads at exons 1 and 2. Chi-squared test was used for statistical analysis. **b** *Msh2*$^{+/+}$ and *Msh2*$^{-/-}$ MEFs were treated as shown in the upper box. Expression of the replicative Polδ and the TLS polymerases η, κ, and ι was analyzed by western blotting. Red arrowheads indicate the signals when each polymerase was highly expressed. Graphs show means ± s.e. ($n = 3$ independent experiments with MEFs prepared from independent fetuses). **c**, **d** *Msh2*$^{+/+}$ and *Msh2*$^{-/-}$ MEFs were treated according to the workflow. γH2AX foci were detected by immunofluorescence (**c**, *n* numbers are indicated in graph). Percentages of γH2AX foci colocalized with 53BP1 foci (means ± s.e.) are indicated in each image (**d**). Bars show means ± s.d. Scale bars, 10 μm. Two-tailed Welch's *t*-test was used for statistical analysis

Next, we investigated point mutations in the DNA-binding domain (DBD) of p53; these are the most common cancer-driver p53 mutations[34]. As expected, these mutations were observed in several independently immortalized subcultures: 12 independent mutations out of 24 independently immortalized *Msh2*$^{-/-}$ MEFs and three independent mutations out of 12 independently immortalized *Msh2*$^{+/+}$ MEFs (Supplementary Fig. 5b). Mutations resulting from G/T-mismatches predominated in *Msh2*$^{-/-}$ MEFs. Thus, the mutations that perturb ARF/p53-module functions in *Msh2*$^{-/-}$ MEFs included base substitutions in the p53 DBD and chromosomal deletion at the *Cdkn2a* gene locus, analogous to those in cancer cells (Supplementary Fig. 5a). Importantly, immortalization of *Msh2*$^{-/-}$ MEFs was blocked when replication stress was reduced (Fig. 2a), implying that those two types of mutations are induced under replication stress, along with MSI.

These results suggest that hypermutation arises under replication stress in *Msh2*$^{-/-}$ MEFs (Fig. 3e). In fact, point mutations in the p53 DBD were also likely induced in this cellular state (Supplementary Fig. 5b). To investigate the hypermutagenic cellular background that arises under replication stress, the involvement of low-fidelity translesion synthesis (TLS) polymerases[35] was investigated after MEFs (P8) were treated as shown in the workflow (Fig. 4b, Top). Even MEFs at P8 expressed PCNA upon the onset of DNA replication, and progressively expressed

TLS polymerases, especially pol η, which coincided with Rad51 expression (Fig. 4b). Thus, expression of low-fidelity pol η was induced at a time when γH2AX foci were accumulated (Fig. 2c), implying that DNA synthesis could occur in the absence of proofreading and MMR. Although TLS polymerases also accumulated in *Msh2*$^{+/+}$ MEFs, hypermutation was not detected (Fig. 3b, c), suggesting that it was suppressed by MMR.

**MMR-dependent repair of replication stress-associated DSBs**. Notably, levels of Rad51 and TLS polymerase η were reduced at 36–48 hr after release of *Msh2*$^{-/-}$ MEFs into serum-containing medium, while those reduction levels were less in *Msh2*$^{+/+}$ MEFs (Fig. 4b). This suggests that replication stress-associated DSBs were effectively repaired in *Msh2*$^{-/-}$ MEFs compared to those in *Msh2*$^{+/+}$ MEFs. Consistent with this, while γH2AX foci mostly colocalized with 53BP1 were formed after 24 h following exposure to growth stimuli in both *Msh2*$^{-/-}$ and *Msh2*$^{+/+}$ MEFs, those at 48 h were present at lower levels in *Msh2*$^{-/-}$ MEFs than in *Msh2*$^{+/+}$ MEFs (Fig. 4c, d). In fact, identical results were also shown in γH2AX (or 53BP1) foci colocalized with BrdU under native conditions (Fig. 2c and Supplementary Fig. 3a). These results suggest that DSBs caused by replication stress were effectively repaired in *Msh2*$^{-/-}$ MEFs, but not in *Msh2*$^{+/+}$ MEFs. Tetraploidization (CIN) arises when replication stress-associated DSBs are not effectively repaired, leading to carryover of DSBs

into mitosis and cytokinesis failure[16,18]. Therefore, the difference in the repair efficiency of DSBs between $Msh2^{-/-}$ and $Msh2^{+/+}$ MEFs might be linked to the relatively likelihood of induction of CIN vs. MSI.

To confirm that CIN was suppressed in the MMR-deficient background, we examined the formation of bi-nuclei and micronuclei, which often arise upon CIN induction[16]. As expected, formation of these nuclei was efficiently suppressed in $Msh2^{-/-}$ MEFs (Supplementary Fig. 6a, b). This result implies that formation of aberrant nuclei associated with CIN was effectively suppressed in the MMR-deficient background. Even in $Msh2^{-/-}$ MEFs, we observed some aberrant nuclei, including bi-nuclear tetraploidy (~10%), which was consistent with our chromosome analyses, in which we observed tetraploidy in two of ten nuclei (Supplementary Fig. 2b). Importantly, those tetraploid MEFs usually did not become predominant (Fig. 1b), implying that tetraploidization is not actively associated with induction of mutations in the ARF/p53 module in $Msh2^{-/-}$ MEFs.

**PARP-mediated DSB repair and MSI induction.** A remaining question is how MSI is induced while CIN is suppressed in MMR-deficient cells. Our previously described results indicate that CIN suppression in $Msh2^{-/-}$ MEFs is associated with the effective repair of replication stress-associated DSBs (Fig. 4b, c). This repair pathway is probably erroneous because MSI (i.e., In/Del mutations in repetitive loci) is induced (Fig. 3f, g). Therefore, we studied the effect of the PARP inhibitor Olaparib, which inhibits DSB repair by microhomology-mediated end joining (MMEJ)[36], because microsatellite loci consisting of repetitive sequences are potential microhomologies to promote erroneous repair. As expected, γH2AX foci induced upon exposure to growth stimuli were not effectively reduced in the presence of Olaparib even in $Msh2^{-/-}$ MEFs (Fig. 5a), in which the majority of γH2AX foci were merged with 53BP1 foci (Fig. 5b). Thus, DSBs formed upon exposure to exogenous growth stimuli are repaired via a PARP-mediated pathway in an MMR-deficient background.

To investigate whether this PARP-mediated repair pathway induces MSI and suppresses CIN under replication stress, $Msh2^{-/-}$ MEFs at P7 were continuously treated with Olaparib (Fig. 5b, left panel). Intriguingly, upon continuous Olaparib treatment, immortalized $Msh2^{-/-}$ MEFs exhibited CIN with tetraploidy, rather than MSI (Fig. 5c). Thus, we conclude that replication stress-associated DSBs induced upon exposure to continuous growth stimuli trigger both CIN and MSI. In an MMR-deficient background, MSI is induced via PARP-mediated erroneous repair and CIN is concomitantly suppressed.

**MSI-associated CPT resistance acquisition.** Our previously described results indicate that the induction of MSI (or CIN) in response to replication stress-associated DSBs is tightly associated with the induction of cancer-driver mutations, leading to clonal expansion of cells with abrogated defense systems. This prompted us to investigate whether MSI induction is associated with the acquisition of drug resistance by MMR-deficient cancer cells, especially to anti-cancer drugs that cause replication stress, such as camptothecin (CPT). MMR-deficient HCT116 cells were treated with CPT such that their survival rate was reduced to 0.1–0.2% as shown in the workflow and compared with MMR-proficient HeLa cells (Fig. 6a). During this process, cells that survived under drug treatment eventually recovered proliferative activity and formed colonies (Supplementary Fig. 7a). Although both cell types exhibited increased drug resistance, MSI was only observed in HCT116 cells (Fig. 6a). Thus, MMR-deficient cancer

cells can acquire resistance to drugs that cause replication stress in association with MSI.

To investigate the repair of replication stress-associated DSBs, replication stress was induced via pulse treatment with hydroxyurea as shown in the workflow (Fig. 6b). As expected, the number of γH2AX foci in HCT116 cells decreased over time and this was blocked by the PARP inhibitor Olaparib, but not by the DNA-PK inhibitor NU7441, which prevents non-homologous end joining (Fig. 6b). In addition, these γH2AX foci were continuously detected in HeLa cells unless MSH2 and MLH1 were depleted (Supplementary Fig. 7b). Thus, replication stress-associated DSBs are effectively repaired via a PARP-mediated pathway in MMR-deficient cancer cells. Together, our findings demonstrate that when MMR-deficient cancer cells are treated with the anti-cancer drug CPT, some cells survive and acquire resistance, in association with MSI induction, and that replication stress-associated DSBs are repaired in these cells via a PARP-mediated pathway.

To further investigate the repair pathway that induces MSI but suppresses CIN, we studied the involvement of Polθ, which also mediates DSB repair via microhomology[37,38]. Indeed, even in HCT116 cells, γH2AX foci induced by replication stress following hydroxyurea (HU) treatment persisted when Polθ was knocked down (Fig. 6c and Supplementary Fig. 7c), supporting the idea that MMEJ is the major pathway for repair of DSBs caused by replication stress in an MMR-deficient background. To address the involvement of Polθ in MSI induction, Polθ-KD HCT116 cells were treated as shown in the workflow in Fig. 6d, and their MSI status was determined and compared with those of negative control NC HCT116 cells transfected with the empty vector. Although both types of cells exhibited elevated resistance, MSI was only observed in NC cells, but not in Polθ-KD cells (Fig. 6d). Together, these results indicate that MSI induction, in association with CIN suppression, is triggered by replication stress-associated DSBs via a repair pathway mediated by both Polθ and PARP, which is likely mediated by MMEJ in an MMR-deficient background.

## Discussion

The current study illustrates a pathway leading to MSI and hypermutation that is triggered by replication stress in an MMR-deficient background (Fig. 7). During this process, MSI is induced via a PARP-mediated repair pathway and CIN is concomitantly suppressed. Hypermutation simultaneously arises in association with the induction of low-fidelity TLS polymerases. Importantly, this leads to clonal expansion of cells harboring ARF/p53-module mutations and that have acquired resistance to the anti-cancer drug CPT. This process can be circumvented by CIN, although CIN induces mutations much less efficiently than MSI. It was recently shown that replication stress arises in association with transcription and collisions of the transcription machinery with replication forks[39] and that common fragile sites (i.e., hotspots of CIN-associated chromosomal abnormalities), which usually encompass repetitive sequences, are enriched in introns of large genes[40,41]. Our results are consistent with these findings; MSI was massively induced in introns as an alternative to CIN (Supplementary Fig. 4h).

Cancer generally develops via multiple steps, each of which involves the clonal evolution of cells with abrogated defense systems[42–44]. This in vitro study suggests that replication stress-triggered MSI/hypermutation (or CIN) could underlie such stepwise progression, at least in cells that have lost ARF/p53-dependent defense systems and acquired resistance to the anti-cancer drug CPT (Fig. 7a). This is conceptually similar to stress-induced mutagenesis[45–47]. Given that most cancers develop with

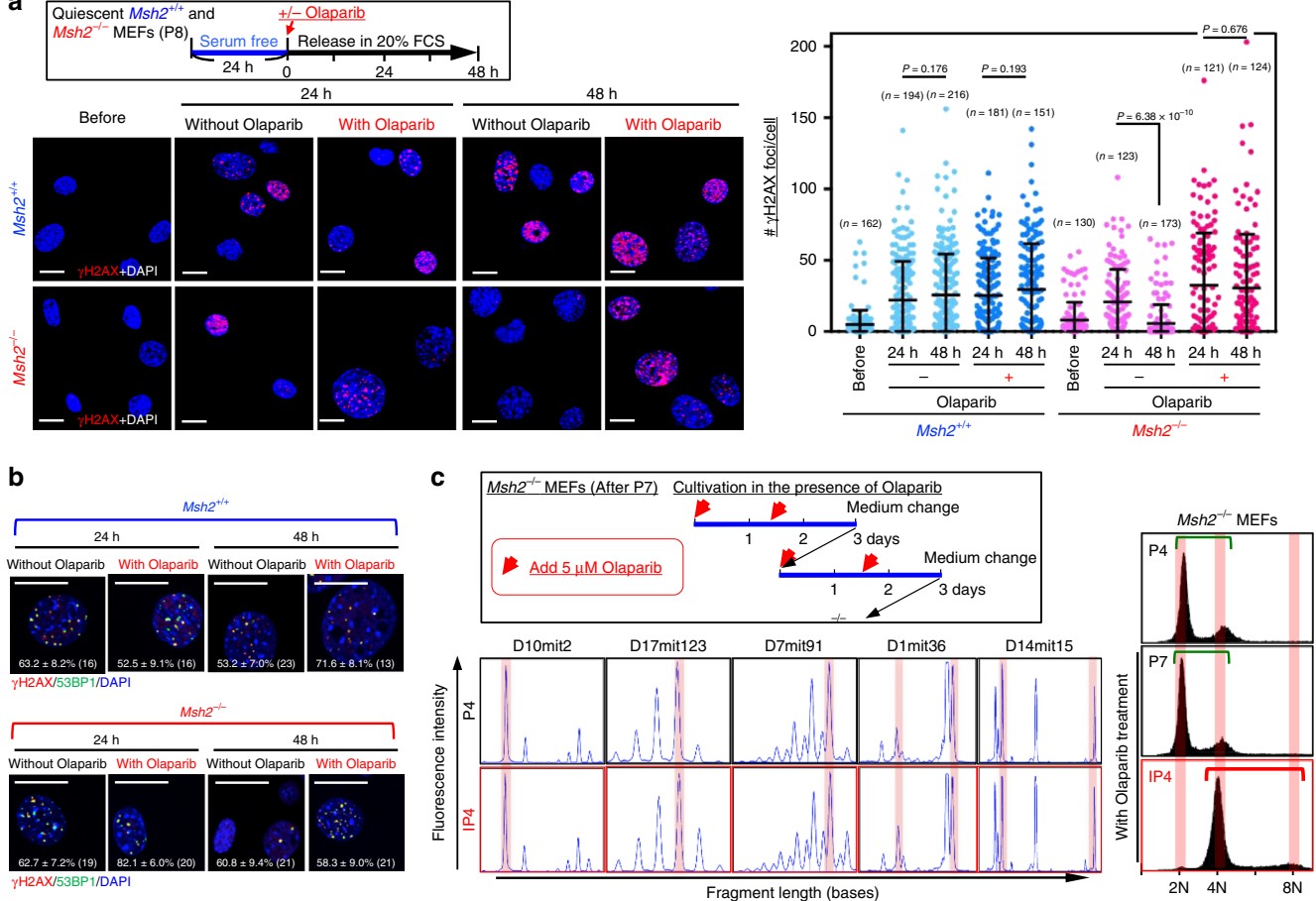

**Fig. 5** PARP mediates repair of replication stress-associated DSBs and MSI induction in $Msh2^{-/-}$ MEFs. **a, b** $Msh2^{+/+}$ and $Msh2^{-/-}$ MEFs were treated according to the workflow (upper box). γH2AX foci (**a** $n$ numbers are indicated in graph) and γH2AX/53BP1 foci (**b**) were detected by immunofluorescence. Percentages of γH2AX foci merged with 53BP1 foci (means ± s.e.) are indicated in each image (**b**). Bars show means ± s.d. Scale bars, 10 μm. Two-tailed Welch's $t$-test was used for statistical analysis. **c** $Msh2^{-/-}$ MEFs were cultivated under the Std-3T3 protocol and continuously treated with Olaparib, and their CIN-induction and MSI statuses were monitored. The MSI status was determined at the indicated loci. The CIN-induction status was determined by flow cytometry. Green and red bars indicate diploidy and tetraploidy, respectively

CIN or MSI, replication stress-triggered MSI/hypermutation (or CIN) could be a cause of such clonal evolution through the induction of clonal expansion of cells mutated in their defense systems.

MMR proteins have multiple functions including MMR[6,8,9] and checkpoint activation in response to certain types of DNA adducts such as O[6]-methyl G[48]. A key question is which function elicits cancer-suppressive effects. MMR-dependent checkpoint activation seems to be important for certain types of chemotherapy but does not elicit anti-cancer effects because MMR-deficient mice with certain types of mutations in which the DNA damage checkpoint response still occurs are predisposed to cancer with MSI[49,50]. Our results are consistent with those of previous studies because we demonstrated that induction of hypermutation in MMR-deficient cells is tightly associated with induction of MSI, induction of mutations in cancer-driver genes, and resulting clonal expansion. Moreover, our results revealed that the risk is particularly elevated under replication stress rather than during canonical replication, illustrating a setting in which MMR plays an important role.

Induction of MSI in MMR-deficient cells occurs together with suppression of CIN. Specifically, replication stress-associated DSBs, which can cause CIN, are repaired via MMEJ. This raises the question of how different repair pathways are selected in MMR-deficient and MMR-proficient backgrounds. Although the underlying mechanism is unclear, it probably involves the formation of different complexes comprising MMR proteins and multiple other repair factors. Indeed, MMR proteins interact with a wide variety of repair factors[51].

It remains unknown how cancer-driver mutations are induced and whether they are avoidable[52,53]. The standard view is that hypermutation arises in MMR-deficient cells due to their inability to correct replication errors[8,23], and is therefore unavoidable. Given that some errors occur during normal replication, we assume that the p53 DBD could be mutated in a few cells among $10^4$ $Msh2^{-/-}$ MEFs at P8, although some should be silent mutations. Under this assumption, cells bearing mutations in both alleles should arise no more often than once in $10^7$ cells. However, the immortalization frequency of $Msh2^{-/-}$ MEFs (~1/ $10^4$) was much higher than this when those MEFs subjected to MSI induction under replication stress (Fig. 4b), illustrating the pronounced effects of replication stress-triggered hypermutation and MSI on the induction of these mutations. Importantly, unlike errors caused during normal replication, replication stress-triggered mutations are potentially preventable by eliminating the source of replication stress, in association with maintenance of genome stability.

MSI and CIN were suppressed during the proliferating state but spontaneously arose when the growth rate slowed, although the cultivation conditions remained unchanged (Fig. 1). These

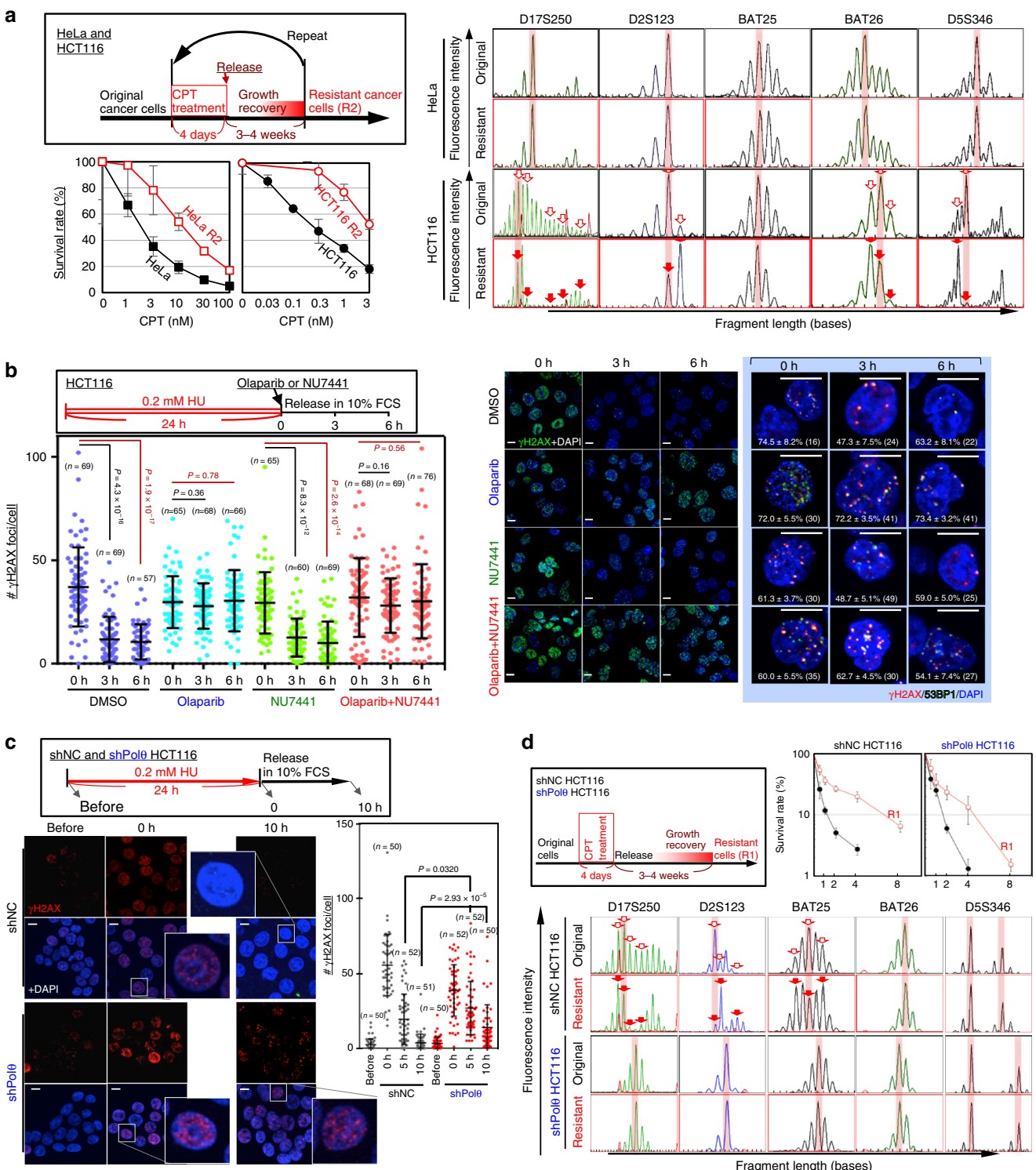

increases in the induction of MSI and CIN probably correlate with the accumulation of unrepairable DSBs in association with aging in vivo and passage in vitro[54]. In fact, normal cells often exhibit deficiencies in the repair of replication stress-associated DSBs after their growth rate slows[18], although they can still repair DSBs directly caused by γ-rays[55]. The reasons for this repair deficiency remain unclear, but might involve the reduced level of H2AX expression in the growth-arrested state;[18,19] this protein is required for genome stability. These factors may explain why the risk of cancers, especially those that develop following genomic instability, increases with age. Our results suggests that the induction of MSI and CIN is associated with the induction of cancer-driver mutations and subsequent clonal expansion, demonstrating the importance of genome stability maintenance when cells become sensitive to replication stress.

## Methods

**Cell culture.** $Msh2^{+/+}$, $Msh2^{+/-}$, and $Msh2^{-/-}$ MEFs were prepared from fetuses of $Msh2^{+/-}$ mice[56] and cultured using a Std-3T3 passage protocol[57] or a modified protocol (tSD-3T3) to generate quiescent MEFs (P8)[18]. To obtain immortalized MEFs, MEFs that reached the growth-arrested state (P8) were maintained with a medium change every 3 days without passaging until they exhibited immortal growth (IP1). HCT116 and HeLa cells were also used[58]. All cells were cultured in

**Fig. 6** PARP mediates MSI induction and CIN suppression. **a** The HCT116 (MMR-deficient) and HeLa (MMR-proficient) cancer cell lines were treated as shown in the upper box to induce resistance to CPT. The original and resulting (R2) cells were subsequently treated with CPT, and their survival efficiencies and MSI induction at the indicated loci were assessed. Red arrows indicate the shifted fragment peaks, i.e., MSI. Graphs show mean survival rates ± s.d. ($n$ = 3 independent experiments with the same cell line). **b** HCT116 cells treated with HU were cultivated in the presence and the absence of Olaparib and NU7441. Thereafter, γH2AX foci or γH2AX/53BP1 foci were monitored by immunofluorescence ($n$ numbers are indicated in graph). Percentages of γH2AX foci merged with 53BP1 foci (means ± s.e.) are indicated in each image ($n$ numbers are in images). Bars show means ± s.d. Scale bars, 10 μm. Two-tailed Welch's $t$-test was used for statistical analysis. **c** HCT116 cells, with or without stable down-regulation of Polθ, were treated with HU (see upper box), and γH2AX accumulation was monitored by immunofluorescence ($n$ numbers are indicated in graph). Bars are shown as mean ± s.d. Scale bars, 10 μm. Two-tailed Welch's $t$-test was used for statistical analysis. **d** HCT116 cells, with or without stable down-regulation of Polθ, were treated as shown in the upper-left box to obtain cells resistant to CPT. The resultant and original cells were treated with CPT, and their survival efficiencies and MSI statuses were assessed. Red arrowheads indicate the shifted fragment peaks, i.e., MSI. Graph shows mean cell numbers ± s.d. ($n$ = 3 independent experiments with three independent clones)

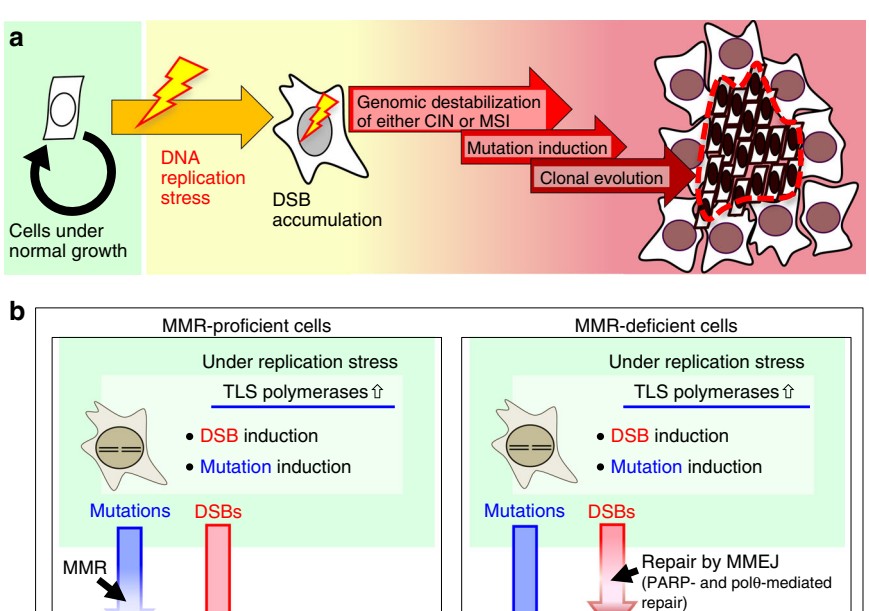

**Fig. 7** Model. **a** Cells that accumulate replication stress-associated DSBs are at a higher risk of genomic destabilization of either CIN or MSI. Genomic destabilization is associated with induction of mutations, leading to clonal evolution of cells with defects in cellular defense systems. **b** CIN is caused when replication stress-associated DSBs are not repairable, whereas MSI arises in association with erroneous DSB repair by MMEJ. Hypermutation is also induced in MMR-deficient cells

Dulbecco's Modified Eagle's Medium (Nakarai) supplemented with 10% (v/v) fetal calf serum (FCS; Gibco).

**MSI status analysis**. MSI statuses during immortalization (MEFs) and resistance acquisition (HCT116 cells) were determined by the change in the fragment length at five independent microsatellite loci: D10mit2 (di-nucleotide repeats), D17mit123 (di-nucleotide repeats), D7mit91 (di-nucleotide repeats), D1mit36 (di-nucleotide repeats), and D14mit15 (di-nucleotide repeats) for MEFs and D17S250 (di-nucleotide repeats), D2S123 (di-nucleotide repeats), BAT25 (mono-nucleotide repeats), BAT26 (mono-nucleotide repeats), and D5S346 (di-nucleotide repeats) for human cancer cells. To analyze the MSI status during MEF immortalization, genomic DNA was prepared from all cultured cells, or a portion thereof if continuous culture was required. Each microsatellite fragment was amplified by PCR and analyzed on a Beckman Coulter CEQ8000 analyzer. The MSI status was judged by the change in fragment length after immortalization of MEFs or resistance

acquisition. The primers, which were conjugated with a fluorescent dye, for each locus were as follows:

D10mit2, [F]CTGCTCACAACCCATTCCTT/GTTCATTTGAGGCACAAGCA

D17mit123, [F]CACAAGGAGGGAGCCTGTAG/CACCGTAAGAGTCTAAT AATAAGGGG

D7mit91, [F]TCTTGCTTGCATACACTCACG/GAGACAAACCGCAGTCTC CT

D1mit36, [F]GAGGAATGTAGAGTCCAACCTGG/TGAATAGATTAAGAG CCTGGAAGC

D14mit15, [F]TTGGCTGCTCACTTGCAG/TTACCCTCCCCATAACTCCC

D17S250, [F]GGAAGAATCAAATAGACAAT/GCTGGCCATATA TATATTTAAACC

D2S123, [F]AAACAGGATGCCTGCCTTTA/GGACTTTCCACCTATGGAC

BAT25, [F]TCGCCTCCAAGAATGTAAGT/TCTGGATTTTAACTATGG CTC

BAT26, [F]TGACTACTTTTGACTTCAGCC/AACCATTCAACATTTTTAA CC

D5S346, [F]ACTCACTCTAGTGATAAATCGGG/AGCAGATAAGACAAG
TATTACTAG

**Chromosomal instability status analyses**. The CIN induction status was determined by flow cytometry after propidium iodide staining (10 μg/ml) in PBS containing 0.1% Triton X-100 and 100 μg/ml RNaseA[48]. For G-band analyses, cells cultivated in the presence of 12.5–25 ng/mL colcemid for 1.5–5 h were trypsinized, collected by centrifugation (1500 rpm for 5 min), suspended in 75 mM KCl (5 mL) and incubated for 20 min Carnoy's solution (acetic acid:methanol = 1:3) (6 mL) was added to the cell suspension before centrifugation (1500 rpm for 5 min). The pellet was washed twice by suspending cells in Carnoy's solution (5 mL) with subsequent centrifugation (1500 rpm for 5 min). The resulting chromosome samples were again suspended in Carnoy's solution (0.1–1 mL) and spread onto glass slides. The slides were incubated at 80 °C for 48 hr, soaked in 0.025% trypsin prepared in phosphate-buffered saline (PBS) (37 °C) for a few seconds, and then washed once in 10% FCS prepared in PBS and once in 5% FCS prepared in PBS. For G-banding, the slides were stained by soaking in 3% Giemsa prepared in PBS for 10 min and subsequently washed with water. Chromosomes were monitored by microscopy (Zeiss Axio Imager Z2).

**Cell biological experiments**. DNA damage was induced by treatment with CPT (Sigma) and HU (Sigma). Olaparib (Selleckchem) and NU7441 (Selleckchem) were also used. Western blotting was performed using antibodies indicated below after blotting onto PVDF membrane[58]. Proteins were transferred to PVDF membranes for 2 or 15 h for detection of ATR.

Cells for immunofluorescence were prepared using primary and secondary antibodies indicated below after 4% paraformaldehyde fixation, permeabilization with 0.1% Triton X-100/PBS, and blocking (2% Goat serum in PBS containing 0.3% Tritin X-100)[58,59]. Immunofluorescence was performed using a confocal laser microscope (Olympus FV10i). γH2AX and 53BP1 foci in each nucleus were automatically counted using the "Find Maxima" function of ImageJ after manual identification of nuclei visualized by DAPI staining. Foci counting was performed under the same conditions in each experiment. EdU staining was performed with Click-iT Plus EdU Imaging Kit (Life Technologies). BrdU staining was performed in the cells administered 10 μM BrdU[60]. Those cells were washed with PBS and pre-extracted (25 mM Hepes, pH 7.4, 50 mM NaCl, 1 mM EDTA, 3 mM MgCl₂, 300 mM sucrose, and 0.5% Triton X-100) for 20 min on ice before fixation with 4% paraformaldehyde and immunostaining. BrdU was detected by immunofluorescence under native conditions on a confocal laser microscope (Zeiss LSM880), or after denaturation with 2 M HCl when indicated.

Survival rates were determined by counting the number of viable cells after CPT treatment, comparing with the numbers before the treatment[58]. Lipofectamine RNAiMax (Life Technologies) was used to transfect siRNAs. The sequences of the top-strand siRNAs targeting MSH2, MLH1, and Polθ were as follows:

MSH2, UCUGCAGAGUGUUGUGCUUTT
MLH1, GCCAUGUGGCUCAUGUUACUTT
Polθ, GCUUCAGUGAUGACUAUCUAGUAAA (siPolθ#1)
Polθ, CAUUCGGGUCUUGGCGGCAACUUCU (siPolθ#2)

The immortalization efficiency was determined after MEFs (P8) were seeded at a density of $1 \times 10^5$ cells/well (6-well plate), $2 \times 10^4$ cells/well (12-well plate), $1 \times 10^3$ cells/well (48-well plate), $2.5 \times 10^3$ cells/well (48-well plate), or $5 \times 10^3$ cells/well (48-well plate).

For resistance acquisition experiments, cells were exposed to CPT under conditions in which the survival rate was reduced to 0.1–0.2% and then released into CPT-free medium until growth was recovered. This procedure was performed once or twice to obtain HCT116 and HeLa R1 or R2 cells.

**Antibodies**. Antibodies against the following proteins and nucleotide were obtained from the indicated suppliers: 53BP1 (PC712, Merck; 1/500), α-tubulin (T6074, Sigma; 1/5000), β-actin (AC-74, Sigma; 1/5000), H3 (MABI0301, MBL; 1/10000), γH2AX (9718, Cell Signaling; 1/400), PCNA (ab29, Abcam; 1/1000), ATR (sc-515173, Santa Cruz; 1/250), phospho-ATR (Thr1989) (GTX128145, GeneTex; 1/500), RPA32 (2208, Cell Signaling; 1/1000), phospho-RPA32 (Ser33) (E-AB-21080, Elabscience; 1/1000), Polδ (ab129498, Abcam; 1/500), Polη[61] (A301–231A, Bethyl; 1/1000), Polι (1/3000)[61], Polκ (generated in rabbit and purified with a Protein G column; 1/3000), Rad51 (8875, Cell Signaling; 1/1000), and BrdU (66241–1-Ig, proteintech; 1/200). Secondary antibodies were obtained from the indicated suppliers: anti-Mouse IgG-HRP (NA931, GE healthcare; 1/5000), anti-Rabbit IgG-HRP (NA934, GE healthcare; 1/5000), anti-Mouse IgG-Alexa Fluor 488 (A-11001, Thermo Fisher; 1/1000), and anti-Rabbit IgG-Alexa Fluor 594 (A-11012, Thermo Fisher; 1/1000).

**Exome analysis**. Whole exome sequencing of $Msh2^{+/+}$ and $Msh2^{-/-}$ MEFs at each stage was performed and analyzed after samples were prepared from $Msh2^{+/+}$ and $Msh2^{-/-}$ MEFs at each stage based on the manufacturer's protocol[62] (Agilent).

1. Sequencing procedures. DNA was extracted from MEFs at each stage. Whole exome capture was accomplished based on liquid-phase hybridization of sonicated genomic DNA (mean length of 150–200 bp) to the bait cRNA library synthesized on magnetic beads (Agilent Technology), according to the manufacturer's protocol.

The SureSelect XT Mouse All Exon Library was used. The captured targets were subjected to massive sequencing using Illumina GAIIx and/or HiSeq 2000 with the pair-end 75–108 bp read option, according to the manufacturer's instructions.

2. Pipeline for data processing. The raw sequence data generated by Illumina GAIIx or HiSeq2000 sequencers were processed through the in-house pipeline constructed for whole exome analysis of paired cancer genomes at the Human Genome Center, Institute of Medical Science, University of Tokyo. The data processing was divided into two steps: 1) generation of a .bam file (http://bio-bwa. sourceforge.net/) for paired $Msh2^{+/+}$ and $Msh2^{-/-}$ samples for each case, and 2) detection of somatic point mutations and indels by comparing .bam files of $Msh2^{+/+}$ and $Msh2^{-/-}$ samples. Candidate somatic mutations were detected through the Genomon-exome pipeline [http://genomon.hgc.jp/exome/][62].

2.1 Generation of .bam files.

2.1.1 Preprocessing. Initially, .fastq files originally generated from Illumina sequencers were converted to .fastq in Sanger format using bcl2fastq Conversion Software v1.8.4 [http://support.illumina.com/downloads/ bcl2fastq_conversion_software_184.html]. PCR adapter sequences contaminating the sequence reads were removed using Cutadapt [https://cutadapt.readthedocs.io/ en/stable/].

2.1.2 Mapping of sequence reads and detection of duplicate reads. Sequenced reads were aligned to the NCBI Mouse Reference Genome Build 37 with BWA (version 0.7.8 and default parameter settings) [http://bio-bwa.sourceforge.net/]. The output was written into a .sam file, which was converted into a .bam file for subsequent calculations via Samtools0.1.18 [http://samtools.sourceforge.net/]. The aligned reads were examined using the MarkDuplicates algorithm from Picard 1.39 [https://broadinstitute.github.io/picard/] to identify molecular duplicates, where a read is considered to be a molecular duplicate if both ends of the pair reads are mapped to identical genomic locations. The detected duplicates were flagged in the .bam file.

2.2 Detection of somatic mutations and indels.

2.2.1 Generation of the .pileup files for $Msh2^{+/+}$ and $Msh2^{-/-}$ MEF data. Before summarizing the base-call data, low-quality reads were eliminated from each .bam file, including those reads with more than five mismatches to the reference sequences or whose mapping quality score was <30. The sequence data in .bam files were then summarized into a .pileup file, which contained the counts of each base call at every nucleotide position in the target sequences. To avoid an excessive number of false-positive findings, the following nucleotide positions were eliminated from further analysis, including those positions at which the depth was less than 10 in either the $Msh2^{+/+}$ or $Msh2^{-/-}$ MEF samples, is adopted as the candidate mutation.

2.2.2 Statistical evaluation of single nucleotide variants (SNVs) and indels. The significance of each candidate mutation was evaluated by Fisher's exact test by enumerating the numbers of the reference base and the candidate SNV in both the $Msh2^{+/+}$ and $Msh2^{-/-}$ MEF samples. Candidate mutations with p-values of less than 0.001 were adopted as provisional candidates for somatic mutations.

**Accession numbers**. The sequencing data obtained by exome analyses were deposited in the DDBJ database (under the accession number DRA005173).

**Mutation analysis of p53 DBD**. To analyze the p53 DBD mutation status in immortalized MEFs, 12 cultures of $Msh2^{+/+}$ MEFs and 24 cultures of $Msh2^{-/-}$ MEFs were independently immortalized. RNA purified using a ReliaPrep RNA Cell Miniprep System (Promega) was reverse-transcribed, and the resulting cDNA was amplified by PCR using the high-fidelity polymerase KOD-plus (Toyobo) for cloning into the TA cloning vector (Takara). The PCR primers were 5′-ACGCTTCTCCGAAGACTGG-3′ (forward) and 5′-GGACGGGATGCA-GAGGCAGT-3′ (reverse). To confirm the mutations, capillary sequencing was performed for multiple clones, each of which was independently amplified by PCR. The primers used for sequencing were 5′-ACAGGACCCTGTCACCGAGA-3′ (forward) and 5′-CGGATCTTGAGGGGTGAAA-3′ (reverse).

**Analyses of mutation-induction status in human cancers**. To identify mutations that perturb the functions of p53 and ARF in human cancers, we analyzed gastric cancer data obtained from The Cancer Genome Atlas using software available at cbioportal.com[63,64]. p53 and Cdkn2a mutations were analyzed in MSI-high and MSS (microsatellite stability)-positive gastric cancers.

To investigate the In/Del status of human colorectal cancers, we analyzed sequence data from ten colorectal cancer patients obtained from The Cancer Genome Atlas[65]. The patients were categorized into two groups, MMR-deficient and MMR-proficient (five in each group). MMR deficiency was identified based on the presence of MMR gene mutations, MSI-high, and an elevated mutation rate, and MMR proficiency was characterized based on the lack of MMR gene mutations, MSS, and a lower mutation rate. In/Del statuses were specifically analyzed and compared between groups.

Information regarding replication timing of exonic regions was obtained from ReplicationDomain[66]. Replication timing in MEFs was obtained from data of wild-type MEFs (accession number: Int65970816), while that in human cancer cells was obtained by averaging two sets of data from HCT116 cells (accession numbers: Int90617792 and Int97243322). The resulting data were divided into two groups,

i.e., early (50%) and late (50%). Accordingly, the replication timing of each In/Del position was categorized as early or late.

**Stable transformant cell constructions**. To obtain the shPolθ vector, shRNA target sequence (shPolθ: 5′-GCTGACCAAGATTTGCTATAT-3′) was cloned into vector pBASi hU6 (TAKARA). HCT116 cells transfected with each vector (shPolθ, and empty control) were selected by treatment with 0.5 μg/mL puromycin for 14 days. Clones were picked and expanded for an additional 14 days, and then Polθ mRNA levels were assessed.

**Mutation number assumption during canonical replication**. The expected numbers of mutations caused by replication errors and detectable by exome analyses were estimated in $Msh2^{-/-}$ MEFs at IP1 and IP28. After fertilization, cells are estimated to divide no more than 48 times up to P8, specifically, 38 times during embryogenesis (12 times during segmentation and 26 times (twice per day) up to 13.5 days) and 10 times during P1–P7 (twice per passage during P1–P3 and once per passage during P4–P7). Mutations induced during early segmentation are detectable at P4 and hence mutations that become detectable at IP1 did not arise during the first three divisions. MEFs expanded from a single immortalized cell had divided ~19 times by IP1, and mutations that arose during the first three divisions could be detected by exome analysis. In summary, MEFs at IP1 could accumulate mutations over 48 or fewer divisions. About 25 base substitutions occur in the total genome per division in each cell during normal growth[67], as estimated in a human organoid model in a MMR-deficient background. The DNA replication and repair machineries are well conserved in mammals; therefore, this rate was used to estimate the expected number of mutations during normal replication. The numbers of mutations caused in exon are expected as about 1.5% of total genome, based on the size ratio of exon/genome. Thus, fewer than 18 base substitutions were expected to be detectable at IP1 in all exons. Although multiple clones could potentially immortalize from $3 \times 10^5$ $Msh2^{-/-}$ MEFs, MEFs that immortalized first became predominant. In exome analyses, most mutations (89% at IP1) were detected in 30–60% reads of their depths (Fig. 3d). This indicates that $Msh2^{-/-}$ MEFs at IP1 with mutations detectable by exome analyses were clones of single immortalized cells.

Cells are expected to divide about 54 times during IP1–IP28 (approximately twice per passage). Mutations caused by replication errors after IP1 and detectable at IP28 could be mainly dependent on the probability of clonal expansion. This can be estimated using the following formula:

$$_{4n}C_m \times (1/4)^m \times (3/4)^{4n-m} \left( m = 1, 2, \ldots, 3\times10^5,\ n = 1, 2, \ldots, 3\times10^5 \right) \quad (1)$$

Where $m$ and $n$ indicate the numbers of cells in a given passage and the preceding passage, respectively. Accordingly, the expected numbers of cells expanded at IP28 from a single cell at each passage can be estimated using the following formula:

$$\sum_{n=1}^{3\times10^5} l_{4n}C_m \times (1/4)^m \times (3/4)^{4n-m} \quad (2)$$

where $l$ is the number of cells that possess the original mutation. To investigate the accumulation of mutations, the expected cell numbers at IP28 were plotted and superimposed for the plots of each passage (Supplementary Fig. 3d). Estimation was performed for $n = 1, 2, \ldots, 50$ and $m = 0, 1, 2, \ldots, 200$, where the passed cell number was $3 \times 10^5$. Mutations are only detectable with exome analysis when cells with each mutation are expanded more than 10%, i.e., $3 \times 10^4$ cells per dish. In our estimation, most cells expanded to 1–50 cells after serial passage (Supplementary Fig. 3e). This indicates that mutations caused by replication errors during IP1–IP28 are hardly detectable by exome analysis.

Unlike mutations caused after IP1, some mutations caused during clonal expansion but not detected at IP1 could become detectable at IP28 after expansion. Immortalized MEFs are expected to divide about 19 times during clonal expansion, in which mutations induced during 1st–3rd divisions could be detected at IP1 and hence are not included as the mutations detected at IP28. The mutation accumulations depend on the expansion situations during IP1–IP28; therefore, numbers of detectable mutations were estimated with several expansion situations (Supplementary Fig. 3f), in which the assumption was under the same expansion situations in all IP28 MEFs. Mutations are detectable when cells with those mutations are expanded more than 10%. Single IP1 cells are expected to be expanded as in Supplementary Fig. 3g. Thus, numbers of detectable mutations at IP28 were estimated about 10, based on the expected expansion status and those expected numbers at each expansion situation.

**Reporting summary**. Further information on research design is available in the Nature Research Reporting Summary linked to this article.

## Data availability
Exome data presented in this study has been deposited in the DDBJ database (under the accession number DRA005173). TCGA datasets of mutations in human tumors are available from The Cancer Genome Atlas[65]. Source data for all Figs and Supplementary

Figs are provided as a Source Data file. All data is available from the corresponding author upon reasonable request.

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

## Acknowledgements

We thank Drs. R. D. Wood, F. Esashi, P. Hsieh, H. Saya, N. Onishi, B. Shiotani, and H. Teraoka for critical discussion of this manuscript; Drs. T. Shibata and F. Hosoda for support with Illumina sequencing; Dr. S. Dobashi for technical support; Mr. Y. Sato for support in immunofluorescence by Zeiss LSM880; Dr. Y. Shiraishi for support in TCGA data analyses. This study was supported by Grant-in-Aid for Scientific Research on Innovative Areas (13H04908), the MEXT/JSPS KAKENHI (20770136), and the National Cancer Center Research and Development Fund (23-C-10).

## Author contributions

Y.Ma., Y.A., A.S., H.F., M.H., and Y.Mi. designed the work, performed experiments, and analyzed the data; K.K., T.S., and S.M. performed the exome experiments and analyzed the data; Y.N. and T.T. prepared samples and interpreted the data; S.K., R.H., and F.H. analyzed and interpreted the data; K.Y. designed and supervised the project, performed experiments, interpreted the data, and wrote the manuscript.

## Additional information

**Competing interests:** The authors declare no competing interests.

