## [Peer Review File · Nature Communications]

Reviewers' comments:

Reviewer #1 (Remarks to the Author):

Replication stress-associated DNA double-strand breaks as a trigger of microsatellite instability, hypermutation, and clonal evolution

DNA Mismatch repair deficient (dMMR) cancers are characterized by highly elevated rates of small insertion/deletion and single base substitution mutations. dMMR cancers occur in many different types of malignancies.

MMR proteins associate with the replication fork by interacting with pcna during the cell cycle. Less is known about MMR mechanisms during G0 in response to double strand breaks (DSBs). This study focuses on immortalization of mouse Msh2^{-/-} MEFs and the origins of MSI/dMMR. Msh2^{-/-} MEFs immortalize by increased MSI small indel mutations, and have fewer structural variant (SV) mutations.

SVs are induced under a Std-3T3 protocol and Msh2^{-/-} MEFs did not immortalize. Given that replication stress-associated DSBs trigger SV, the authors suggest that replication stress is involved in MSI induction. γ H2AX foci were observed in proliferating MEFs that repair efficiency of DSBs differs according to the MMR status. Bulk whole exome sequencing was performed of MEFs with proliferation, and results consistent with a single cell origin of colonies. Bulk total read depths of most mutations at later passages were lower than those of mutations detected initially, consistent with clonal expansion. Consistent with less selective pressure on non-coding mutations, high rates of noncoding mutations were also seen in Msh2^{-/-} MEFs. Interestingly, expression of low-fidelity polymerases was highly induced when replication stress-associated DSBs accumulated, similar to TCGA observed POLD1 and POLE mutations in dMMR tumors. They used Olaparib to provide evidence that DSBs formed upon exposure to exogenous growth stimuli are repaired via a PARP-mediated pathway in an MMR-deficient background. They then treated HCT116 cells with CPT and showed that MMR-deficient cancer cells can acquire resistance to drugs that cause replication stress in association with MSI.

This is a detailed study of dMMR mutagenesis in Msh2^{-/-} MEFs, with some follow up in HCT116 cells. The study raises some interesting questions regarding replication stress. For Nature Communications, it would be useful for the authors to follow up these studies with analyses of dMMR tumors in different public resources, such as TCGA or COSMIC. They could focus on early vs late replicating genomic regions, which affects MMR repair, transcribed vs untranscribed, coding vs non-coding regions to elucidate whether or not their findings in a single mouse MEF line are supported in dMMR patient tumors. Or, to perform scDNA sequencing in dMMR tumor or evaluate paired primary-metastases and perform clonality analyses to test their hypotheses. These latter experiments would help move the field forward and strengthen the generalizability of their findings.

More detail on statistical techniques and thresholds used in this study would also be helpful.

Reviewer #2 (Remarks to the Author):

The authors studied the immortalization process of MMR proficient (WT MEF) and MMR-deficient MEF (Msh2 KO MEF). Both cell lines showed high levels of γ H2AX particularly in EdU positive cells, which is likely related to replication stress (RS). The authors conclude that RS associated DSB cause chromosomal instability (in this case, tetraploidy) in WT MEFs, whereas MSI and suppression of CIN is found in Msh2 KO MEFs.

Overall the study is potentially interesting but the conclusions are rather preliminary.

Some of the data are of good quality, but others should be substantially improved. I think that the interpretation of the data and the conclusions obtained are in, some cases, overestimated.

Major points:

-It is clear that Msh2 KO cells present high levels of gH2AX foci and increased MSI status. Although it is likely that increased RS/DSBs could give rise to increased MSI, formal proofs for a causal effect are missing.

-gH2AX foci counting is used as marker of RS and DSBs across the different sections of the study. This is a generic marker of DNA damage and can be unspecific. I would recommend to use additional markers, such as 53bp1 (more specific for DSB) or native BrdU (more specific for RS). In addition, it is not clear how the foci were counted. It should be indicated. Counting gH2AX foci is not trivial and usually requires some automatization. The quantification of the total gH2AX signal would also be informative.

-In several figures it is indicated that 3 biologically independent experiments were performed. What does this exactly mean? Has the experiment been performed 3 times with the same cell line or 3 independent cell lines (primary or immortalized) have been analyzed? Due to the variability between cell lines, particularly after immortalization, I believe that analyzing several clones is very recommended.

-I think that the western blot shown in Figure 4b is not of enough quality. The loading control, B-actin, is very variable and I think that most of the changes in polymerases described by the authors could just be reflecting the loading differences. More replicates of this experiment, and using several WT and KO cell lines should be performed to obtain convincing conclusions.

Reviewer #3 (Remarks to the Author):

In this study, the authors use a combination of different immortalization protocols combined with different analyses such as analysis of accumulation of mutations, chromosome abnormalities (CIN; mostly gain and loss of chromosomes), mutations in key target genes, histone H2Ax and Rad51 foci among others to analyze the processes that might underlie the accumulation of high levels of mutations in mismatch repair (MMR) defective msh2^{-/-} cells. The key result is that the authors convincingly show a correlation between the induction of replication stress and both the increased accumulation of mutations and well as increased co-H2Ax-Rad51 foci in msh2^{-/-} cells and increased accumulation of chromosome abnormalities and well as increased co-H2Ax-Rad51 foci in msh2^{+/-} and msh2^{+/+} cells. They also provide some insight into what pathways might promote mutations and suppress chromosome abnormalities in msh2^{-/-} cells. With the caveats listed below, I think these basic results are well established and that the involvement of replication stress in induction of mutations in MMR defective cells is an important result of broad general interest in cancer biology.

My first major concern about this study, which I think can be easily addressed, is poor quantitation of data. In discussing individual experiments, the authors provide virtually no indication of for example, how many cells were examined in individual experiments and what % co-localization of foci is seen or even how many foci are seen. In the mutation analysis they simply list more mutations as the key result but never provide an analysis indicating that the frequency (or rate) of mutations goes up. Similarly, in the analysis of chromosome abnormalities in msh2^{-/-} cells, they never state what % of cells analyzed have such abnormalities or the number of abnormalities per cell for comparison to the MMR proficient case. It is possible that the writing style obscures these

results but more likely the authors are taking the modern tact of overwhelming the reader with experiments presented in a qualitative way rather than emphasizing quantitative proof. I am confident the authors have the data they need to present without doing additional experiments.

The second key concern lies with whether the induction of histone H2Ax Rad51 foci represents just double strand breaks (DSBs) and whether these are in fact a precursor to the increased mutations/MSI in *msh2*^{-/-} cells. The more likely explanation is that these are a broader DNA damage marker than just DSBs that indicates induction of replication stress. To prove that they are DSBs, the authors need to use a more specific marker such as 53BP1 foci or Mre11 foci. Even if they are just DSBs, the authors don't have any direct evidence that DSBs underlie increased mutations as compared to being the damage that underlies CIN in MMR proficient cells and whose more appropriate repair in *msh2*^{-/-} cells accounts for reduced CIN vs increased MSI. In this regard, I find the PARP pathway experiments interesting but don't see how they show that it is DSBs that are repaired resulting in increased mutations and MSI. Basically, the authors would be more realistic to propose that it is replication stress and possibly replication stress associated DNA damage or fork stalling and repair/restart that leads to mutations/MSI. This is an important conclusion that advances the field significantly. Further molecular analysis of the damaged DNA and its repair, which should be a separate study (this study has plenty of work in it), would be required to establish an exact mechanism.

Overall, if properly revised and more conservatively stated, I would be very supportive of publication.

Specific Comments

P2, L3. I think we know how MMR defects contribute to cancer development - through increased mutation rates leading to increased accumulation of mutations in key genes.

P3, L12-15. I believe there are reports of MMR defective tumor cell lines that also have CIN; see Abdel-Rahman et al PNAS 2001 and Douglas et al Cancer Res 2004. In this regard, it is known that MMR suppresses recombination between divergent DNA sequences and hence MMR defects could contribute to increased CIN. I agree that the MSI phenotype is more prominent, most likely because of the large increase in mutation rates that occur due to MMR defects.

P3, L18. In addition, I would suggest the authors reference Hombauer et al Cell 2011 and Hombauer et al Science 2011 as these papers provide the definitive case for replication coupling of MMR.

P4, L18, 19. It is not clear what about the data in Sup Fig 2b shows CIN in *msh2*^{-/-} cells. Please explain.

Also, in Sup Fig 2a, two analyses are shown, with the left showing only a small number of chromosomal alterations and the right showing many. What is the difference between the two? What is unclear about this analysis and that in Fig 1, as well as Sup Fig 1 is how many nuclei were examined in each case and hence what % of *msh2*^{-/-} cells show MSI only and what % show CIN (also what are the %s for *msh2*^{+/+} and *+/+* cells)?

P5, L10 and Fig 2. Did the authors examine CIN in the *msh2*^{-/-} cells or just MSI?

P5, L11. The authors need to make it clear that the tSD-3T3 protocol does not involve continuous exposure to growth stimuli.

P5, L12. ...as an alternative to CIN in *msh2*^{-/-} MEFs that undergo immortalization during continuous exposure to growth stimuli.

P5, L17-19. How many cells were analyzed and what % co-localization was seen?

A major issue with all of the H2Ax and Rad51 foci experiments is these are not specific DSB markers but rather are broader markers of DNA damage. To more specifically detect DSBs the authors need to use better DSB-specific foci markers such as 53BP1 or Mre11.

P6, L12 to P7, L19. It is not entirely clear to me how these data show that the *msh2*^{-/-} MEFs have hyper mutation. Most of the analysis, for example see Fig 3b, reports number of mutations found. While it is true that for example, the authors report an apparent increase in mutations at IP1 and IP28 for *msh2*^{-/-} vs *msh2*^{+/+}, this doesn't mean that the mutation rates or frequencies are different between the two analyses. Shouldn't the authors report the frequency of mutations found per standard number of kb or reads in these types of analyses to justify that hypermutation occurs. I agree that there are mutation spectra changes.

P8, L11, 12. Shouldn't the authors normalize the number of reads obtained to a gene that does not appear to be deleted.

Fig. 4c. There is no upper box showing the treatment scheme.

P8, L17-20 and Fig. 5b. The labeling of this figure is not clear and the text does not indicate how many of the clones analyzed had a p53 mutation. Does each line in the figure indicate the mutations found in 1 independent clone? Or is each mutation an independent mutation from an independent clone. The authors state in the methods section how many independent clones were analyzed but the data presentation obscures the key results relating to how many clones has a mutation (or mutations) and how many independent mutations were found.

P10, L12. Is this really the only repair pathway PARP inhibition inhibits?

P10, L14, 15. The foci are not repaired. It is the underlying damage that is not repaired.

P14, L6. What type of microsatellite is present at each locus? Mononucleotide repeat? Dinucleotide repeat? Other? Please indicate.

Reviewer #1 (Remarks to the Author):

Replication stress-associated DNA double-strand breaks as a trigger of microsatellite instability, hypermutation, and clonal evolution.

DNA Mismatch repair deficient (dMMR) cancers are characterized by highly elevated rates of small insertion/deletion and single base substitution mutations. dMMR cancers occur in many different types of malignancies.

MMR proteins associate with the replication fork by interacting with pcna during the cell cycle. Less is known about MMR mechanisms during G0 in response to double strand breaks (DSBs). This study focuses on immortalization of mouse Msh2^{-/-} MEFs and the origins of MSI/dMMR. Msh2^{-/-} MEFs immortalize by increased MSI small indel mutations, and have fewer structural variant (SV) mutations.

SVs are induced under a Std-3T3 protocol and Msh2^{-/-} MEFs did not immortalize. Given that replication stress-associated DSBs trigger SV, the authors suggest that replication stress is involved in MSI induction. γ H2AX foci were observed in proliferating MEFs that repair efficiency of DSBs differs according to the MMR status. Bulk whole exome sequencing was performed of MEFs with proliferation, and results consistent with a single cell origin of colonies. Bulk total read depths of most mutations at later passages were lower than those of mutations detected initially, consistent with clonal expansion. Consistent with less selective pressure on non-coding mutations, high rates of noncoding mutations were also seen in Msh2^{-/-} MEFs. Interestingly, expression of low-fidelity polymerases was highly induced when replication stress-associated DSBs accumulated, similar to TCGA observed POLD1 and POLE mutations in dMMR tumors. They used Olaparib to provide evidence that DSBs formed upon exposure to exogenous growth stimuli are repaired via a PARP-mediated pathway in an MMR-deficient background. They then treated HCT116 cells with CPT and showed that MMR-deficient cancer cells can acquire resistance to drugs that cause replication stress in association with MSI.

This is a detailed study of dMMR mutagenesis in Msh2^{-/-} MEFs, with some follow up in HCT116 cells. The study raises some interesting questions regarding replication stress. For Nature Communications, it would be useful for the authors to follow up these studies with analyses of dMMR tumors in different public resources, such as TCGA or COSMIC. They could focus on early vs late replicating genomic regions, which affects MMR repair, transcribed vs untranscribed, coding vs non-coding regions to elucidate whether or not their findings in a single mouse MEF line are supported in dMMR patient tumors. Or, to perform scDNA sequencing in dMMR tumor or evaluate paired primary-metastases and perform clonality analyses to test their hypotheses. These latter experiments would help move the field forward and strengthen the generalizability of their findings.

More detail on statistical techniques and thresholds used in this study would also be helpful.

Response: We greatly appreciate the reviewer's comments and suggestions. As the reviewer suggested, we added analyses of the relevant data obtained from TCGA. Specifically, we analyzed the repeat numbers and repeat lengths of InDels specifically induced in exons of colorectal cancer cells, and then compared them between MMR-proficient and -deficient

backgrounds, which are the most relevant to our current study. The results of those analyses now appear in Supplemental Figure 3a–c. Consistent with our current results, InDels are specifically induced at repetitive loci in an MMR-deficient background, but not in an MMR-proficient background. The repeat numbers peak at around 4–8 bases, similar to our observations in immortalized *Msh2*^{-/-} MEFs (Fig. 3f, g). Thus, our results regarding MSI induction strongly resemble observations made in cancer cells.

Importantly, these analyses revealed the differences in the types of InDels caused in immortalized MEFs vs. human colorectal cancers. Most of the mutations in *Msh2*^{-/-} MEFs were one-base deletions, whereas insertions are also quite common in MMR-deficient colorectal cancers. As the reviewer suggested, it would be interesting to focus on early- vs late-replicating genomic regions. Although we were not able to effectively analyze publicly available data in that manner, due to the focus of this particular study, we are currently working on late-replicating loci in a follow-up project, which will be published elsewhere. We are also enthusiastic to analyze data from public resources as a part of this ongoing project.

The reviewer also made the important suggestion that we sequence dMMR tumor samples. We thank the reviewer for this comment; we plan to include such analyses in a future study. Although we did not add a new sequencing study of dMMR tumor samples to the current manuscript, this should not be an issue regarding the publication of this particular study: the editor has informed us that we are not necessarily expected to provide additional scDNA-seq or new patient-derived sequencing data in our revision of this manuscript.

Between these data and another analysis of public data from TCGA, in the latter case focusing on deletion status at *Cdkn2a* gene locus and point mutations in p53 in human cancers (Supple Fig. 5a), we believe that our study is relevant to the development of human cancers.

Reviewer #2 (Remarks to the Author):

The authors studied the immortalization process of MMR proficient (WT MEF) and MMR-deficient MEF (Msh2 KO MEF). Both cell lines showed high levels of γ H2AX particularly in EdU positive cells, which is likely related to replication stress (RS). The authors conclude that RS associated DSB cause chromosomal instability (in this case, tetraploidy) in WT MEFs, whereas MSI and suppression of CIN is found in Msh2 KO MEFs.

Overall the study is potentially interesting but the conclusions are rather preliminary.

Some of the data are of good quality, but others should be substantially improved.

I think that the interpretation of the data and the conclusions obtained are in, some cases, overestimated.

Response: We greatly appreciate the reviewer's helpful and important comments. Based on these recommendations, we have endeavored to improve and clarify the manuscript.

Major points:

-It is clear that Msh2 KO cells present high levels of γ H2AX foci and increased MSI status. Although it is likely that increased RS/DSBs could give rise to increased MSI, formal proofs for a causal effect are missing.

Response: This is a very important comment. First, as the reviewer suggested in the following comment, we performed experiments using additional markers: 53BP1 (which is more specific for DSB) and native BrdU (which is more specific for RS). Importantly, the majority of both γ H2AX and 53BP1 foci induced after growth acceleration were colocalized with native BrdU foci (Fig. 2c and Supple Fig. S3a). Together with the colocalization of γ H2AX and 53BP1, demonstrated in subsequent experiments, we believe that this observation provides a strong support for our conclusions regarding RS-associated DSB accumulation and its association with genomic instability. In addition, we also detected up-regulation of p-RPA and p-ATR (Supp. Fig. S3b), providing additional support for the presence of replication stress.

We also performed additional experiments to show colocalization of foci of γ H2AX and 53BP1, which supports the link between RS and DSBs in this cellular state. Together with the data showing the reduction in γ H2AX foci and the resultant suppression of MSI under tSD-3T3 conditions, we believe that our study has been significantly improved. To avoid any overstatement regarding RS/DSBs and the associated MSI induction, we carefully rewrote the relevant portions of the manuscript.

- γ H2AX foci counting is used as marker of RS and DSBs across the different sections of the study. This is a generic marker of DNA damage and can be unspecific. I would recommend to use additional markers, such as 53bp1 (more specific for DSB) or native BrdU (more specific for RS).

In addition, it is not clear how the foci were counted. It should be indicated. Counting γ H2AX foci is not

trivial and usually requires some automatization. The quantification of the total γ H2AX signal would also be informative.

Response: As the reviewer suggested, we performed experiments using additional markers: 53BP1 foci for DSBs and native BrdU for RS. To show the data from the native BrdU study, we added new Fig. 2c and New Supple Fig. S3a. The data show colocalization of native BrdU with γ H2AX and 53BP1, supporting the idea that these foci represent RS-associated DSBs. In addition, we also added results showing colocalization of γ H2AX with 53BP1 in new Figures 2d, 4c, 5a, and Supplemental Figure 6c. Those results were basically as expected, but they provide important support for our argument regarding replication stress and the associated DSBs.

In addition, in the Methods section, we added details about the method for counting foci.

-In several figures it is indicated that 3 biologically independent experiments were performed.

What does this exactly mean? Has the experiment been performed 3 times with the same cell line or 3 independent cell lines (primary or immortalized) have been analyzed?

Due to the variability between cell lines, particularly after immortalization, I believe that analyzing several clones is very recommended.

Response: We thank the reviewer for this comment. To produce the data for Figures 1a and 3b, we performed these experiments three times, using MEFs prepared from three independent fetuses; this is analogous to analyzing separate clones. This information has been added to the appropriate figure legends.

-I think that the western blot shown in Figure 4b is not of enough quality.

The loading control, B-actin, is very variable and I think that most of the changes in polymerases described by the authors could just be reflecting the loading differences.

More replicates of this experiment, and using several WT and KO cell lines should be performed to obtain convincing conclusions.

Response: We re-performed these experiments, this time using alpha-tubulin and H3 for the loading control. We believe that the data are now acceptable.

Reviewer #3 (Remarks to the Author):

In this study, the authors use a combination of different immortalization protocols combined with different analyses such as analysis of accumulation of mutations, chromosome abnormalities (CIN; mostly gain and loss of chromosomes), mutations in key target genes, histone H2Ax and Rad51 foci among others to analyze the processes that might underlie the accumulation of high levels of mutations in mismatch repair (MMR) defective *msh2*^{-/-} cells. The key result is that the authors convincingly show a correlation between the induction of replication stress and both the increased accumulation of mutations and well as increased co-H2Ax-Rad51 foci in *msh2*^{-/-} cells and increased accumulation of chromosome abnormalities and well as increased co-H2Ax-Rad51 foci in *msh2*^{+/-} and *msh2*^{+/+} cells. They also provide some insight into what pathways might promote mutations and suppress chromosome abnormalities in *msh2*^{-/-} cells. With the caveats listed below, I think these basic results are well established and that the involvement of replication stress in induction of mutations in MMR defective cells is an important result of broad general interest in cancer biology.

Response: We thank the reviewer for these critical and supportive comments. We believe that our manuscript has been greatly improved by addressing the reviewer's concerns.

My first major concern about this study, which I think can be easily addressed, is poor quantitation of data. In discussing individual experiments, the authors provide virtually no indication of for example, how many cells were examined in individual experiments and what % co-localization of foci is seen or even how many foci are seen. In the mutation analysis they simply list more mutations as the key result but never provide an analysis indicating that the frequency (or rate) of mutations goes up. Similarly, in the analysis of chromosome abnormalities in *msh2*^{-/-} cells, they never state what % of cells analyzed have such abnormalities or the number of abnormalities per cell for comparison to the MMR proficient case. Its possibly that the writing style obscures these results but more likely the authors are taking the modern tact of overwhelming the reader with experiments presented in a qualitative way rather than emphasizing quantitative proof. I am confident the authors have the data they need to present without doing additional experiments.

Response: We revised the figures and figure legends and provided information about cell numbers, percent of co-localized foci, and foci numbers per cell in each figure or legend; some of the data plots actually show “the foci numbers per cell”. In addition, we carefully re-wrote the detailed methods for foci counting. In the mutation analysis, we added information about the mutation rate in Fig. 2b and 2c (please see the right-side y-axis). Based on those data, we also improved the text accordingly. I believe that these figures are now acceptable.

Regarding the data “% of chromosome abnormalities per cell”, we have added this information to Supp. Fig. S2c-d as a score of apparent chromosome abnormalities. However, I would still explain about these analyses, as following.

In fact, I agree completely that it would be much better if we could perform analyses similar to those in recent powerful studies, as the reviewer suggested. However, the cells we are using create an obstacle to that approach. Although MEFs are generally suitable for many types of studies, including this one, it is not possible to perform overwhelming chromosome-karyotyping analyses in these cells because, unlike human chromosomes, mouse chromosomes usually do not spread clearly and are not well characterized. In human chromosomes, we can clearly identify abnormalities based on banding patterns. By contrast, mouse chromosomes are more difficult to analyze, and it is harder to detect chromosomal translocations. Therefore, for these analyses, we only analyzed those chromosomes by karyotyping from ten nuclei to obtain representative images. We now show all the results in Supp. Fig. 2a–b with their apparent chromosomal-abnormality scores (Supplementary Fig. 2c-d). Although the range of possible analysis was limited, we could still see abnormal chromosomal features, illustrating that chromosomal abnormalities are induced even in MMR-deficient backgrounds, but these levels are lower than in an MMR-proficient background.

Although our karyotyping data are not overwhelming, it is fortunate that MMR-proficient MEFs immortalize with tetraploidy, which is easily detectable by flow cytometry, as shown in Figures 1 and 5. We understand that these data differ from those observed by karyotyping, i.e., rates of chromosomal abnormality. However, we could still distinguish massive CIN-associated tetraploidization in association with immortalization.

As the reviewer pointed out below in a specific comment, in our karyotyping analyses we observed some tetraploidy with some aberrant chromosomes even in *Msh2*^{-/-} MEFs. Specifically, tetraploidy was observed in two of ten chromosome spreads. This result is somewhat consistent with our measurement of the accumulation rate of bi-nuclear tetraploid cells; about 10% of MEFs developed bi-nuclear tetraploidy, even in an *Msh2*^{-/-} background in the senescent state (Supplementary Fig. 6). However, tetraploid cells never predominated among immortalized *Msh2*^{-/-} MEFs, suggesting that such tetraploidization does not actively contribute to the development of immortality (i.e., induction of mutation in the ARF/p53 module) in *Msh2*^{-/-} MEFs, in contrast to the MMR-proficient background.

The second key concern lies with whether the induction of histone H2Ax Rad51 foci represents just double strand breaks (DSBs) and whether these are in fact a precursor to the increased mutations/MSI in *msh2*^{-/-} cells. The more likely explanation is that these are a broader DNA damage marker than just DSBs that indicates induction of replication stress. To prove that they are DSBs, the authors need to use a more specific marker such as 53BP1 foci or Mre11 foci. Even if they are just DSBs, the authors don't have any direct evidence that DSBs underlie increased mutations as compared to being the damage that underlies CIN in MMR proficient cells and whose more appropriate repair in *msh2*^{-/-} cells accounts for reduced CIN

vs increased MSI. In this regard, I find the PARP pathway experiments interesting but don't see how they show that it is DSBs that are repaired resulting in increased mutations and MSI. Basically, the authors would be more realistic to propose that it is replication stress and possibly replication stress associated DNA damage or fork stalling and repair/restart that leads to mutations/MSI. This is an important conclusion that advances the field significantly. Further molecular analysis of the damaged DNA and its repair, which should be a separate study (this study has plenty of work in it), would be required to establish an exact mechanism.

Overall, if properly revised and more conservatively stated, I would be very supportive of publication.

Response: We agree with the reviewer. The problem that we had in the previous version was related to a poor characterization of DSBs and replication stress. This issue was also raised by Reviewer 2. As suggested, we performed additional experiments with 53BP1 (more specific for DSB) and native BrdU (more specific for RS when colocalization of γ H2AX and 53BP1 foci are observed). As shown in the revised figures, the majority of those γ H2AX foci colocalized with 53BP1 foci, supporting the idea that they represent DSBs. In addition, those γ H2AX and 53BP1 foci were mostly colocalized with native BrdU foci, supporting the idea that they were associated with replication stress. Together with our additional data, we believe that our findings and arguments regarding RS/DSBs and the associated induction of mutation and MSI induction are more convincing. Moreover, we believe the results showing the involvement of the PARP-mediated repair pathway are also more informative. The PARP-dependence is further supported by our new data showing dependence on PolQ (Fig. 6c-d and Supple Fig. S7).

As suggested by the reviewer, we carefully re-wrote the text to avoid any over-interpretation or overstatement. We believe that the relevant sections of the paper are now acceptable.

Specific Comments

P2, L3. I think we know how MMR defects contribute to cancer development - through increased mutation rates leading to increased accumulation of mutations in key genes.

Response: We carefully rewrote this section.

P3, L12-15. I believe there are reports of MMR defective tumor cell lines that also have CIN; see Abdel-Rahman et al PNAS 2001 and Douglas et al Cancer Res 2004. In this regard, it is known that MMR suppresses recombination between divergent DNA sequences and hence MMR defects could contribute to increased CIN. I agree that the MSI phenotype is more prominent, most likely because of the large increase in mutation rates that occur due to MMR defects.

Response: We thank the reviewer for this comment. We re-wrote the Introduction, and we feel that this section is now clearer and fairly cites previous work.

P3, L18. In addition, I would suggest the authors reference Hombauer et al Cell 2011 and Hombauer et al Science 2011 as these papers provide the definitive case for replication coupling of MMR.

Response: Thank you for this comment. We cited those papers.

P4, L18, 19. It is not clear what about the data in Sup Fig 2b shows CIN in *msh2*^{-/-} cells. Please explain.

Response: We carefully rewrote this part of the figure legend to clarify what is actually shown. Together with the new data shown in Supple Fig. S2a-d, we believe that these arguments are now convincing.

Also, in Sup Fig 2a, two analyses are shown, with the left showing only a small number of chromosomal alterations and the right showing many. What is the difference between the two? What is unclear about this analysis and that in Fig 1, as well as Sup Fig 1 is how many nuclei were examined in each case and hence what % of *msh2*^{-/-} cells show MSI only and what % show CIN (also what are the %s for *msh2*^{+/+} and *+/+* cells)?

Response: This is a criticism related to the first major comment raised by the reviewer. As explained above, we analyzed ten nuclei for both *Msh2*^{-/-} and *Msh2*^{+/+} MEFs. Here, all immortalized *Msh2*^{+/+} MEFs exhibited tetraploidy with multiple additional chromosomal abnormalities, as expected. We also observed chromosomal abnormalities even in *Msh2*^{-/-} MEFs, in which those abnormalities are less abundant than in *Msh2*^{+/+} MEFs.

Although those immortalized MEFs must be clones of single immortalized MEFs harboring mutations in the ARF/p53 module, their apparent chromosomal abnormalities were not uniform. These observations indicate that even in the *Msh2*^{-/-} background, MEFs in this state are susceptible to CIN-associated genomic destabilization in addition to MSI induction. The frequency of chromosomal abnormalities in *Msh2*^{-/-} MEFs is still significantly lower than that in *Msh2*^{+/+} MEFs. This is consistent with our repair results (Figs. 4c-d and 5a-b and Supple Fig 6), showing that efficient DSB repair is associated with CIN suppression in *Msh2*^{-/-} MEFs. Importantly, although the chromosomal abnormalities induced in *Msh2*^{-/-} MEFs included tetraploidy, the tetraploid fraction never became predominant, as shown by our flow cytometry analyses. In addition, quite a high population (four of ten nuclei) of *Msh2*^{-/-} MEFs acquired the immortality without any visible chromosomal abnormalities, suggesting that those CIN associated genomic instability is not required for the immortalization of *Msh2*^{-/-} MEFs. However, our results also suggest that chromosomal abnormalities often cause cancer-driver mutations, e.g., through deletion induction in *Cdkn2a* gene locus in *Msh2*^{-/-} cells (in contrast to *Msh2*^{+/+} MEFs, tetraploidization is probably not required in *Msh2*^{-/-} cells).

As explained in our reply above about the quality of karyotyping analysis in MEFs, the analytical methods available to us were limited. However, we could still see

CIN-associated abnormalities, and differences in their abundance, in *Msh2*^{-/-} and *Msh2*^{+/+} MEFs.

P5, L10 and Fig 2. Did the authors examine CIN in the *msh2*^{-/-} cells or just MSI?

Response: In this experiment, we also looked at CIN status by flow cytometry to determine whether abnormal ploidy was induced. This result is now described in the revised text and Figure 2a. Because the cells at this state were largely quiescent, they rarely undergo M phase. Therefore, it is not possible to obtain sufficient numbers of M-phase chromosome spreads.

P5, L11. The authors need to make it clear that the tSD-3T3 protocol does not involve continuous exposure to growth stimuli.

Response: We added the requested information to the appropriate sections of the manuscript.

P5, L12. ...as an alternative to CIN in *msh2*^{-/-} MEFs that undergo immortalization during continuous exposure to growth stimuli.

Response: We thank the review for this comment. We added this information to the text.

P5, L17-19. How many cells were analyzed and what % co-localization was seen?

Response: Information about cell number and percent co-localization is now indicated in the figure legends for γ H2AX/53BP1, γ H2AX/native BrdU, and 53BP1/native BrdU.

A major issue with all of the H2Ax and Rad51 foci experiments is these are not specific DSB markers but rather are broader markers of DNA damage. To more specifically detect DSBs the authors need to use better DSB-specific foci markers such as 53BP1 or Mre11.

Response: We performed additional experiments with 53BP1. These data have been replaced with Rad51/ γ H2AX foci data.

P6, L12 to P7, L19. It is not entirely clear to me how these data show that the *msh2*^{-/-} MEFs have hyper mutation. Most of the analysis, for example see Fig 3b, reports number of mutations found. While it is true that for example, the authors report an apparent increase in mutations at IP1 and IP28 for *msh2*^{-/-} vs *msh2*^{+/+}, this doesn't mean that the mutation rates or frequencies are different between the two analyses. Shouldn't the authors report the frequency of mutations found per standard number of kb or reads in these types of analyses to justify that hypermutation occurs. I agree that there are mutation spectra changes.

Response: We agree. We added the mutation rate induced for immortalized *Msh2*^{-/-} vs *Msh2*^{+/+} MEFs. Please see the right-side y-axis. This is a more direct comparison to show how much mutations are induced.

P8, L11, 12. Shouldn't the authors normalize the number of reads obtained to a gene that does not appear to be deleted.

Response: We added the information with fold changes of relative read numbers at exons 1alpha and 2, which are normalized by total read numbers in whole exome analyses. Please see new Fig. 4a (right-side graph).

Fig. 4c. There is no upper box showing the treatment scheme.

Response: We apologize for omitting the upper box. We added a box showing the treatment scheme to the revised figure.

P8, L17-20 and Fig. 5b. The labeling of this figure is not clear and the text does not indicate how many of the clones analyzed had a p53 mutation. Does each line in the figure indicate the mutations found in 1 independent clone? Or is each mutation an independent mutation from an independent clone. The authors state in the methods section how many independent clones were analyzed but the data presentation obscures the key results relating to how many clones has a mutation (or mutations) and how many independent mutations were found.

Response: We believe the reviewer is referring to Supplementary Fig. 5b and P8, L17–20. We regret our insufficient explanation of those experiments. We have revised the text of the Results and Methods sections, as well as the relevant figure legends.

We analyzed 24 independent clones of *Msh2*^{-/-} MEFs and 12 clones of *Msh2*^{+/+} MEFs. The mutations observed in each clone are indicated in Fig. 5b. In total, we detected 12 independent *p53* mutations in 24 *Msh2*^{-/-} clones, and three independent *p53* mutations in 12 *Msh2*^{+/+} clones. This information has been added to the Methods section and the appropriate figure legend.

P10, L12. Is this really the only repair pathway PARP inhibition inhibits?

Response: We thank the reviewer for raising this point; we believe this is an important issue. We made identical observations in PolQ-KD. PolQ mediates microhomology-mediated end joining (MMEJ), but does not participate in HR or NHEJ; we performed this experiment to determine whether any of the other MMEJ factors were involved. We have added these data in the revised version. Please see Fig. 6 and Supp. Fig. 7. We carefully re-wrote the related part of the text.

We did not add these data to the previous version simply because we thought the story became too complex when so much information was included.

We believe that the additional data about PolQ-KD makes the manuscript more informative because it strengthens our identification of the repair pathway that mediates MSI.

Together, our data suggest that MMEJ is the likeliest repair pathway.

P10, L14, 15. The foci are not repaired. It is the underlying damage that is not repaired.

Response: We thank the reviewer for this comment. This has been revised.

P14, L6. What type of microsatellite is present at each locus? Mononucleotide repeat? Dinucleotide repeat? Other? Please indicate.

Response: We identified both mono-nucleotide and di-nucleotide repeats. This information is now provided in the Methods section.

Reviewers' comments:

Reviewer #1 (Remarks to the Author):

In the original manuscript, because all findings are based on Msh2^{-/-} MEFs and HCT116 cells passaged for more than a decade, I recommended for the authors to provide evidence that their work can be generalized beyond MEFs, and whether similar mechanisms are shared during MMR tumorigenesis. One way to do this would be to analyze different existing studies. Early and late replicating genomic regions have been identified for multiple different tumor types, as well as additional features such as chromatin organization and transcribed vs untranscribed regions. Thus, one way to delineate specifically whether these findings are prominent in dMMR patient tumors would be to evaluate dMMR tumors in different public resources, such as TCGA or COSMIC. An alternative would include performing scDNA sequencing in dMMR tumors, and another to evaluate paired primary-metastases with more contemporary clonality dendrogram analyses to test their hypotheses.

In response, the authors analyzed the repeat numbers and repeat lengths of InDels specifically induced in exons of colorectal cancer cells, and then compared them between MMR-proficient and -deficient backgrounds. This may show that TCGA contains dMMR tumors, but in an era of well delineated mutation signatures, does not address whether the specific issues of early vs late replication, and other related features, from their MEF studies are relevant to dMMR tumorigenesis, or whether these are MEF limited mechanisms. While they do state that they "are currently working on late-replicating loci in a follow-up project, which will be published elsewhere," the bottom line is that the significance of their major findings remains unclear and reduce the impact of this study.

Reviewer #2 (Remarks to the Author):

I am concerned about the new results presented by the authors in response to my previous comments.

Although they tried to address my comments, I believe that the new data presented are not of enough quality. In particular, the 53bp1 stainings in Figures 2b, 4d, 5b and 6b does not show a foci pattern. Therefore, it is not clear to me how the quantification of colocalizing foci with gH2AX was performed. On the other hand, the BrdU staining in Figure 2 is very weak and probably unspecific. Positive and negative controls should be included to confirm the specificity of the signal.

Finally, I must say that I am surprised that after repeating the western blots of polymerases presented in Figure 4b, the current results are completely different and the authors do not explain the reason for those differences.

Reviewer #3 (Remarks to the Author):

In this revised paper, the authors have added considerable new experimental data and significantly revised the manuscript to address my concerns. Overall, I find the study to be well done and that it adds significantly to the field. It does raise some provocative questions that I hope future studies will address as these are beyond the scope of the present study. I do have a few minor points that I believe should be addressed, but which do not require re-review.

1 - On P3, L68 the authors state that "This is generally thought to increase the risk of cancer driver mutations,.....". This is a true statement but shouldn't it be referenced by at least citing a review or two. I also note that Edelmann published 2 papers about separation-of-function

mutations (Cancer Cell, PNAS) that showed it was the mutator phenotype and not DNA damage response defect caused by MMR defects that resulted in increased cancer development.

2 - In Figure 3B the authors state that they are measuring mutation rates. This is not correct as they are not measuring mutations per cell division. Rather they are measuring mutation frequencies as they are measuring mutations per some number of bases sequenced. This is an important difference. Thus the authors need to change the labeling and legends to indicate frequency.

3 - I don't think the authors say enough about the suppression of CIN in MMR deficient cells and how they think this occurs. On P13, L318 to P14, L324 they make an interesting point. Based on this, for example, is it possible that the defect in the suppression of recombination between divergent sequences cause by MMR defects allows more DSBs to be repaired by microhomology mediated end joining resulting in increased MSI and decreased CIN. I think more comment is needed if only to point out that this is an interesting unsolved question raised by the work presented.

Reviewer #1 (Remarks to the Author):

In the original manuscript, because all findings are based on Msh2^{-/-} MEFs and HCT116 cells passaged for more than a decade, I recommended for the authors to provide evidence that their work can be generalized beyond MEFs, and whether similar mechanisms are shared during MMR tumorigenesis. One way to do this would be to analyze different existing studies. Early and late replicating genomic regions have been identified for multiple different tumor types, as well as additional features such as chromatin organization and transcribed vs untranscribed regions. Thus, one way to delineate specifically whether these findings are prominent in dMMR patient tumors would be to evaluate dMMR tumors in different public resources, such as TCGA or COSMIC. An alternative would include performing scDNA sequencing in dMMR tumors, and another to evaluate paired primary-metastases with more contemporary clonality dendrogram analyses to test their hypotheses.

In response, the authors analyzed the repeat numbers and repeat lengths of InDels specifically induced in exons of colorectal cancer cells, and then compared them between MMR-proficient and –deficient backgrounds. This may show that TCGA contains dMMR tumors, but in an era of well delineated mutation signatures, does not address whether the specific issues of early vs late replication, and other related features, from their MEF studies are relevant to dMMR tumorigenesis, or whether these are MEF limited mechanisms. While they do state that they “are currently working on late-replicating loci in a follow-up project, which will be published elsewhere,” the bottom line is that the significance of their major findings remains unclear and reduce the impact of this study.

Response: We greatly appreciate the reviewer’s comment. We have analyzed whether InDels obtained from the TCGA are induced in loci replicating in early or late S phase and compared them with those observed in immortalized MEFs. Please see new Supplementary Figure 4d. These analyses support the relevance of InDels induced in MEFs with those induced in human cancer cells. In fact, InDels induced in both immortalized MEFs and human cancer cells were similar with regard to specific induction in repetitive loci, repeat length, and biased induction in loci replicating in early S phase. As the reviewer pointed out, it is important to determine the relevance of our results to human cancer development. We thank the reviewer for this suggestion.

Reviewer #2 (Remarks to the Author):

I am concerned about the new results presented by the authors in response to my previous comments. Although they tried to address my comments, I believe that the new data presented are not of enough quality. In particular, the 53bp1 stainings in Figures 2b, 4d, 5b and 6b does not show a foci pattern. Therefore, it is not clear to me how the quantification of colocalizing foci with gH2AX was

performed. On the other hand, the BrdU staining in Figure 2 is very weak and probably unspecific. Positive and negative controls should be included to confirm the specificity of the signal.

Finally, I must say that I am surprised that after repeating the western blots of polymerases presented in Figure 4b, the current results are completely different and the authors do not explain the reason for those differences.

Response: We thank the reviewer for this comment. As the reviewer pointed out, these images were not of sufficient quality. We have very carefully repeated the experiments in multiple conditions and analyzed the data. In particular, BrdU staining was performed with several controls. In fact, it took us while. Although the conclusion is unchanged, we believe that these revised images are much more convincing. As the reviewer commented, non-specific background signals were high in the previous images. Please see revised Fig.2c and Supple Fig. 3a and new Supple Fig.3b. In addition, we have provided new images of merged 53BP1 and gammaH2AX foci. 53BP1 is expressed even in nuclei without foci and also forms foci at DSB sites. Therefore, background pan-nuclear signals were usually observed, but merged foci were also clearly detected (please see the revised images). We hope that the reviewer feels that these revised images are acceptable.

We apologize for not explaining why the polymerase results differed from the previous data. Unfortunately, we could not easily reproduce the previous results; therefore, we carefully repeated the western blot experiments several times using two internal controls (alpha-tubulin and histone H3). Consequently, we are very confident about the accuracy of these results, at least with the current lot of MEFs. There may be multiple reasons why the current and previous results differed. First, as the reviewer pointed out, expression of beta-actin, which was used as the internal control in the previous version, was uneven and unstable. Second, primary MEFs prepared from mouse embryos (13.5 days) successively become senescent and quiescent during passage and subsequently acquire immortality. The timings of these transitions and associated sensitivities to exogenous growth stimuli often slightly differ between lots. Therefore, we often inevitably observed differences in the timings of senescence induction and immortality acquisition as well as sensitivities to growth-acceleration stress. Finally, these cellular changes are affected by the culture conditions, such as the lot of FCS and its preparation, i.e., heat-inactivation step. In fact, experiments presented in the previous version were performed 3–4 years ago; therefore, the lots of MEFs and FCS differed between the revised and initial experiments.

We believe that these reasons explain the differences observed. However, TLS polymerase expression was still associated with growth acceleration and induction of

replication stress in the revised experiments. We apologize again for not providing this explanation in the previous version.

Reviewer #3 (Remarks to the Author):

In this revised paper, the authors have added considerable new experimental data and significantly revised the manuscript to address my concerns. Overall, I find the study to be well done and that it adds significantly to the field. It does raise some provocative questions that I hope future studies will address as these are beyond the scope of the present study. I do have a few minor points that I believe should be addressed, but which do not require re-review.

Response: We appreciate the reviewer's helpful comments, which have helped to greatly improve our manuscript.

1 - On P3, L68 the authors state that "This is generally thought to increase the risk of cancer driver mutations,.....". This is a true statement but shouldn't it be referenced by at least citing a review or two. I also note that Edelmann published 2 papers about separation-of-function mutations (Cancer Cell, PNAS) that showed it was the mutator phenotype and not DNA damage response defect caused by MMR defects that resulted in increased cancer development.

Response: We have added the references. We agree that the two papers published by Edelmann regarding separation-of-function mutations are important. Accordingly, we have cited both studies in the revised manuscript. Please see the third paragraph of the Discussion section.

2 - In Figure 3B the authors state that they are measuring mutation rates. This is not correct as they are not measuring mutations per cell division. Rather they are measuring mutation frequencies as they are measuring mutations per some number of bases sequenced. This is an important difference. Thus the authors need to change the labeling and legends to indicate frequency.

Response: We thank the reviewer for this comment and have corrected the labeling in Figure 3b and 3c.

3 - I don't think the authors say enough about the suppression of CIN in MMR deficient cells and how they think this occurs. On P13, L318 to P14, L324 they make an interesting point. Based on this, for example, is it possible that the defect in the suppression of recombination between divergent sequences cause by MMR defects allows more DSBs to be repaired by microhomology mediated end joining resulting in increased MSI and decreased CIN. I think more comment is needed if only to point out that this is an interesting unsolved question raised by the work presented.

Response: We have rewritten the text regarding CIN suppression and MSI induction in MMR-deficient cells. However, as the reviewer pointed out, this issue is an unanswered question raised by the current study. We have clarified this in the Discussion section.

We believe that the underlying mechanism probably involves the formation of different complexes with repair factors. Indeed, MMR proteins interact with multiple repair factors such as components of the BASC complex (Wang et al., *Genes Dev.* 14, 2000). Although the underlying mechanism is unclear, such differences in complex formation probably affect which downstream repair pathway is selected. Please see the fourth paragraph of the Discussion section.

Reviewers' comments:

Reviewer #1 (Remarks to the Author):

In this revision, the authors were requested to re-analyze TCGA MMR-deficient and MMR-proficient tumors and confirm that the loci are similar to those in immortalized MEFs. The authors have added a new Supplementary Figure 4D showing In/del mutations and state that they are similar with regard to repetitive loci, length and early S phase replication.

The new data are fairly light in detail and to my view are not entirely convincing to test their hypotheses on clonal evolution. Rather, they have shown that there is evidence of mutations that occur in early S phase replication, which may or may not relate to their hypothesis on clonal evolution.

Reviewer #2 (Remarks to the Author):

I acknowledge that the authors have performed an important effort to respond to the concerns raised by the reviewers and have addressed most of them.

However, I am still concerned about the results presented in the western blot in Figure 4b.

On one hand, the results presented in the second version are completely different to the first version of the manuscript. The authors argue that this is due to the different lot of MEFs used.

I believe that if the differences are not consistent between different lots of MEFs, is very difficult to interpret the relevance of these observation. Furthermore, I don't see a clear correspondence between the western blot shown and the quantification shown in the figure. According to the authors 3 biologically independent experiments were done. Could the authors provide the images of the 3 blots and the details of the quantifications performed? Does "3 biologically different experiments" mean 3 different MEF lines for each genotype? Finally, I believe that the text does not describe properly what is shown in the graph 4b. The authors claim that "low-fidelity polymerases are highly induced along with Rad51", whereas the graph shows a modest and variable induction for the different polymerases.

The criticisms raised by the reviewers basically concern two issues, namely, use of TCGA data and what can be concluded from the analysis (Reviewer #1) and the worrying inconsistencies observed with different lines of MEFs (Reviewer #2).

Response to the criticism of Reviewer #1

We feel that it is difficult to fully address the comment of the reviewer, but have attempted to demonstrate the relevance of our study to human cancer cells. In addition, we are a little confused about how the reviewer would like us to address his/her comment. Moreover, we feel that this comment likely arose from our insufficient explanation of the analyses in the previous revision.

We agree that it is very important to determine the relevance of our findings to human cancer and appreciate the reviewer's useful suggestion to use publicly available data from TCGA and COSMIC. The reviewer suggested that we investigate early vs. late replicating genomic regions, transcribed vs. untranscribed regions, and coding vs. non-coding regions in a comment on our initial manuscript (NCOMMS-18-20923-T) and early vs. late replicating genomic regions, chromatin organization, and transcribed vs. untranscribed regions in a comment on our first revised manuscript (NCOMMS-18-20923A). However, these investigations are not straightforward because mutation data available in TCGA and COSMIC are limited to exons and are not directly linked to replication timing or chromatin organization. In addition, the reviewer requested that we determine the relevance of our analyses to human cancer cells; therefore, we must compare MEF data and human cancer data. We carefully considered how to address the issues raised by the reviewer in the first and second revisions. During the second revision, we analyzed the location of early vs. late replicating genomic regions using publicly available data from TCGA/COSMIC and our exome data from MEFs. We could only analyze early vs. late replicating genomic regions when comparing human cancer data and MEF data. Although these analyses were limited to InDels in our previous revision (Supplementary Fig. 4d), they

have now been expanded to exonic base substitutions (new Supplementary Fig. 4e). These new data also support the relevance of our findings to human cancers.

We would like to apologize for our insufficient explanations of these analyses and the limitations in the previous revision.

Together with the data shown in Supplementary Figure 4a–d and new Supplementary Figure 4e, we feel that our current dataset demonstrates the relevance of our findings to human cancer, including the specific induction of InDels in repetitive loci, the repeat length, the biased induction of InDels in loci replicating in early S phase, and the biased induction of base-substitution mutations in loci replicating in early S phase. In addition, the specific induction of InDels in repetitive loci was only observed in a MMR-deficient background, consistent with the general knowledge that MSI is specifically induced in MMR-deficient human cancer cells. This supports the relevance of this study to MMR-deficient human cancers. We hope that the reviewer agrees.

Although the reviewer commented that our results are “...not convincing to test their hypotheses on clonal evolution...” in his/her appraisal of the second revised manuscript, we feel that we invited this criticism because of our insufficient response in the previous revision. We believe that the reviewer wants us to expand our findings to human cancer, as much as possible. We have addressed this criticism to the best of our ability by incorporating new data and highlighting the relevance of our findings to human cancer. We carefully re-wrote the text at this section with adding the relevance explanation (Please see page 8).

Regarding “...clonal evolution to show with analyzing TCGA/COSMIC data and/or with additional scDNA Seq...” in human cancers, we feel that this issue cannot currently be addressed even by performing additional NGS analyses of transcribed vs. untranscribed regions, coding vs. non-coding regions, and chromatin organization. This is because all possible NGS analyses of human cancer mutations only investigate the genomic (mutation) states induced in the resulting cancer cells.

Therefore, we cannot tell how and when they are induced or whether they are associated with genomic destabilization. A model system, such as MMR-deficient MEFs, is required for such studies.

In this manuscript, we attempted to elucidate the mechanisms underlying genomic destabilization, mutation induction, and clonal evolution, especially in those correlations including the mechanistic process. We demonstrated genomic destabilization-associated mutation induction, which is triggered by replication stress-associated DSBs and leads to clonal evolution of cells harboring mutations in cancer-driver genes. None of the reviewers had any criticism of this primary conclusion; therefore, we believe that it is convincing, although some criticisms have been addressed in Figure 4b. Although we cannot currently investigate clonal evolution during human cancer development, this study describes the mechanisms by which MSI is induced as well as the associated induction of mutations. In addition, we have investigated the relevance of our findings to human cancer in response to the reviewer's criticism. We appreciate the reviewer's comments.

Response to the criticism of Reviewer #2

We have attached three independent sets of data with quantification. The reviewer was worried about whether the western blot data were correctly quantified; therefore, we have re-quantified the data very carefully. We found inappropriate quantification in some bands, although many of these are still within the margin of error. The major reason for these differences was imprecise background subtraction, as the reviewer also know those quantifications are sometimes tricky. We deeply apologize for those not enough careful quantifications and greatly appreciate the reviewer's criticism, which provided us with the opportunity to improve the quantifications and thereby avoid raising unnecessary doubt about our results. We believe that at least TLS polymerase η is fairly expressed in these experiments.

Regarding the criticism of "...the differences are not consistent between different lots of MEFs, is very difficult to interpret the relevance of these observation", we agree to the reviewer for the importance of this issue. Use of MEFs prepared from independent fetuses is probably a part of reasons to cause relatively higher error levels compared to many other studies done by general cancer lines. Importantly, even with such MEFs prepared from independent fetuses, we still always observed Pol η expression along with Rad51; therefore, we are at least confident in this Pol η expression. We carefully rewrote the text at this section (Please see page 9). We believe that these data are now acceptable.

Here, "...three biologically different experiments..." means three independent experiments with MEFs prepared from independent fetuses, as described in the revised figure legend. We thank the reviewer for this criticism.